# A Scalable Approach for Mapper via Efficient Spatial Search

**Luca Simi** *lucasimi90@gmail.com*

**Reviewed on OpenReview:** *https://openreview.net/forum?id=lTX4bYREAZ*

## Abstract

Topological Data Analysis (TDA) is a branch of applied mathematics that studies the shape of high dimensional datasets using ideas from algebraic topology. The Mapper algorithm is a widely used tool in Topological Data Analysis, used for uncovering hidden structures in complex data. However, existing implementations often rely on naive and inefficient methods for constructing the open covers that Mapper is based on, leading to performance issues, especially with large, high-dimensional datasets. In this study, we introduce a novel, more scalable method for constructing open covers for Mapper, leveraging techniques from computational geometry. Our approach significantly enhances efficiency, improving Mapper's performance for large high-dimensional data. We will present theoretical insights into our method and demonstrate its effectiveness through experimental evaluations on well-known datasets, showcasing substantial improvements in visualization quality and computational performance. We implemented our method in a new Python library called *tda-mapper*, which is freely available at `https://github.com/lucasimi/tda-mapper-python`, providing a powerful tool for TDA practitioners and researchers.

## 1 Introduction

In recent years, Topological Data Analysis (TDA) has gained significant traction in the field of data science due to its ability to extract valuable insights from complex datasets. TDA uses topological methods that are resilient to noise and dimensionality, making it a robust mathematical framework for data analysis. A well-known technique in TDA is the *Mapper algorithm*. Mapper provides a visual representation of data in the form of a graph, called *Mapper graph*, enabling easy exploration and interpretation. Unlike conventional algorithms, such as *clustering algorithms* or *Principal Component Analysis (PCA)*, Mapper excels at visualizing data by preserving their connected components, making it very effective for shape analysis and pattern discovery. The effectiveness of Mapper was initially demonstrated in the analysis of medical data, as showcased in the pioneering work by Singh et al. (2007) and in later works by Carlsson (2009; 2014). Since then, Mapper has proven to be a versatile and powerful tool for data exploration, capable of uncovering hidden patterns even in high-dimensional datasets.

Data exploration is an iterative and interactive process that requires continuous fine-tuning and adjustments to extract meaningful insights. Consequently, software for Mapper must prioritize low running times and minimal memory usage to encourage adoption and be practically useful. The original description of Mapper (Singh et al., 2007) includes what has now become a standard approach, involving the construction of an *open cover* made of overlapping hyperrectangles, also known as *standard cubical cover*. Currently, researchers and developers have access to several established open-source libraries for Mapper. However, these libraries work well with low dimensional lenses, but their approach is often inefficient in higher dimensions. On one hand the standard cubical cover is often assembled from open covers obtained in lower dimensional projections, and this makes the number of steps grow exponentially with the dimension, which is computationally unfeasible. On the other hand, points in dimension $k$ can fall in the intersection of up to $2^k$ hypercubes, resulting in complex Mapper graphs that are hard to explore, and often have more connected components than points are in the dataset. These crucial points has been consistently overlooked and neglected, reinforcing the misconception that Mapper is inefficient with high-dimensional data. Motivated by these limitations, recent advancements in the field have led to the development of a wide family of *Mapper-type* algorithms, each proposing a distinct

adaptation of the original concept. For instance, *Ball Mapper* (Dłotko, 2019) and *Mapper on Ball Mapper* (Paweł Dłotko & Sazdanovic, 2024) construct the open cover by creating an $\epsilon$-net (Gonzalez, 1985), adopting open balls centered in the points of the dataset instead of evenly-spaced hyperrectangles. Additionally, specialized variations like *NeuMapper* (Geniesse et al., 2022), designed specifically for neuroscience data, adopt a more complex approach. This method, partially inspired by Ball Mapper, employs an intrinsic metric derived from *reciprocal kNN*. These adaptations all shift towards changing the way open covers are built, improving computational performance, but giving away the control on the overlap of the open sets. While this is often acceptable, there are cases where using a cubical cover with uniform overlap is beneficial, especially given its foundational role in many Mapper-related results. For instance, it's possible to estimate optimal parameters for the standard cubical cover (Carrière et al., 2018), minimizing the need for time-consuming manual fine-tuning.

In this work, we introduce a novel and more efficient approach to computing Mapper-type algorithms, leveraging concepts from computational geometry, aimed at solving the problems of currently available implementations. Our method uses a greedy adaptation of $\epsilon$-net (Gonzalez, 1985; Dłotko, 2019), called *proximity-net*, to construct a subcover of the standard cubical cover, preserving the evenly overlapping open sets of the standard cubical cover, as defined in the original Mapper implementation. Moreover, we show how we can improve the overall efficiency of Mapper by adopting specialized data structures for spatial search like *metric trees* (Clarkson, 2006; Brin, 1995; Yianilos, 1993; Uhlmann, 1991). We also provide a theoretical analysis on the complexity of building cubical covers, obtaining an upper bound that explicitly incorporates the *doubling dimension* of the dataset (Krauthgamer & Lee, 2004). We present theoretical insights into our method, supported by experimental evaluations on well-known datasets, highlighting significant improvements in running time compared to the standard approach. Additionally, we introduce our open-source library, *tda-mapper* (Simi, 2024), available at `https://github.com/lucasimi/tda-mapper-python`. To the best of our knowledge, it is the only library implementing this approach. Finally, we compare our method with existing software libraries for Mapper, including *Kepler Mapper* (van Veen et al., 2019) and *giotto-tda* (Tauzin et al., 2021). Performance tests demonstrate the advantages of our method in terms of scalability and efficiency, underscoring its potential for large-scale applications.

## 2 Preliminaries

This section introduces the fundamental definitions and notations necessary for the remainder of this work. While many of these concepts are foundational in topology and computational geometry, and are widely available, they are included here for completeness. The concept of $\epsilon$-net, introduced in Gonzalez (1985), is widely used in computational geometry (see also Clarkson (2006)), and is central for the contributions of this work. This notion has been previously used by Dłotko (2019) in the context of TDA as the foundation for a variant of Mapper called *Ball Mapper*. The notion of $\epsilon$-net is also closely related to the *doubling dimension* of a metric space, that was introduced for the first time in Assouad (1983) and later in Gupta et al. (2003). Compared to other definitions in literature, the doubling dimension offers a practical measure of intrinsic dimensionality that remains valuable for discrete finite sets. As we will see later, we will leverage the doubling dimension to estimate the complexity of our proposed approach, in the form of Theorem 3.

**Definition 1.** Let $X$ be a topological space. A *pseudo-metric* on $X$ is a map $d\colon X \times X \to \mathbb{R}$ satisfying:

- $d(x, y) \geq 0$ for all $x, y \in X$, with $d(x, x) = 0$;

- *Symmetry*: $d(x_1, x_2) = d(x_2, x_1)$ for all $x_1, x_2 \in X$;

- *Triangle inequality*: $d(x_1, x_3) \leq d(x_1, x_2) + d(x_2, x_3)$ for all $x_1, x_2, x_3 \in X$.

If $d(x, y) = 0$ implies $x = y$, then $d$ is a metric. An open ball of center $p \in X$ and radius $\epsilon > 0$ is defined as $B_d(p, \epsilon) = \{x \in X \mid d(p, x) < \epsilon\}$.

**Definition 2.** Let $f\colon X \to Y$ be a map and $d$ a pseudo-metric on $Y$. The *pullback* of $d$ under $f$ is the pseudo-metric $f^*d$ on $X$, defined by:

$$(f^*d)(u, v) = d(f(u), f(v)) \quad \text{for all } u, v \in X.$$

**Definition 3.** Let $(X, d)$ be a pseudo-metric space. An $\epsilon$-*net* of $X$ is a subset $N \subseteq X$ such that:

- $d(x, y) \geq \epsilon$ for all $x, y \in N$, $x \neq y$;

- For every $x \in X$, there exists $y \in N$ with $d(x, y) < \epsilon$.

It's possible to construct an $\epsilon$-net using a greedy procedure, as reported in Algorithm 1: we start with an empty set $N$ and at each step we add a point to $N$ that it's further than $\epsilon$ from $N$ itself. At last, when no point is left to pick, we end up with $N$ which is an $\epsilon$-net by construction.

---
**Algorithm 1** Greedy construction of an $\epsilon$-net

---
**Require:** A pseudo-metric space $(X, d)$ and $\epsilon > 0$
**Ensure:** An $\epsilon$-net $N \subseteq X$
  $N \leftarrow \emptyset$
  **while** $d(X \setminus N, N) > 0$ **do**
    Take $p \in X \setminus N$ maximizing $d(p, N)$
    $N \leftarrow N \cup \{p\}$
  **end while**
  **return** $N$

---

**Definition 4.** Let $(X, d)$ be a pseudo-metric space. The *doubling measure* of $X$ is the smallest $\lambda > 0$ such that every ball in $X$ can be covered by $\lambda$ balls of half the radius. The *doubling dimension* of $X$ is defined as $\dim(X) = \log_2 \lambda$.

**Proposition 1.** *The doubling dimension satisfies the following two properties:*

- *Let $X$ and $Y$ be pseudo-metric spaces. If $X \subseteq Y$, then $\dim(X) \leq \dim(Y)$.*

- *For any $p \geq 1$ the vector spaces $\mathbb{R}^k$ with the $L_p$-metric satisfy $\dim(\mathbb{R}^k) = \mathcal{O}(k)$.*

**Proposition 2.** *Let $(X, d)$ be a pseudo-metric space and $N$ an $\epsilon$-net for $X$. Then for any ball $B(p, R)$ in $X$,*

$$|N \cap B(p, R)| = \mathcal{O}\left((R/\epsilon)^{\dim(X)}\right).$$

The time complexity of Algorithm 1 is proportional to the cardinality of the $\epsilon$-net. Therefore, as a consequence of Proposition 2, is $\mathcal{O}\left((R/\epsilon)^{\dim(X)}\right)$.

## 2.1 Mapper algorithm

In this subsection, we provide a concise overview of *Mapper*, based on its original formulation (Singh et al., 2007) (see Figure 1 for an example). Mapper operates on a dataset $X$ and its output is determined by the following steps:

1. Let $f$ be a *lens*, defined as any continuous map $f \colon X \to Y$, where $Y$ is a parameter space. Common choices for the lens $f$ include *statistics* of any order, *projections*, *entropy*, *density*, *eccentricity*, and more.

2. Next, we proceed by constructing an *open cover* for $f(X)$. In other words, we create a collection $\{U_\alpha\}_\alpha$ of open sets such that their union *covers* the entire image $f(X)$, i.e., $f(X) = \bigcup_\alpha U_\alpha$. It is important to note that the sets in this open cover may intersect with one another, and they inherit their topology from the space $Y$.

3. For each element $U_\alpha$ in the selected cover, we define $V_\alpha$ as the preimage of $U_\alpha$ under the function $f$. It is clear that the collection $\{V_\alpha\}_\alpha$ forms an open cover of $X$. Next, we proceed by applying a user-specified *clustering algorithm*, in order to partition each open set $V_\alpha$ into a disjoint union of clusters, denoted as $V_\alpha = \amalg_\beta C_{\alpha, \beta}$. The resulting family $\{C_{\alpha, \beta}\}_{\alpha, \beta}$ is referred to as a *refined open cover* for $X$.

4. We construct the *Mapper graph* as the undirected graph $G = (V, E)$ defined by the following rule: the set $V$ contains a vertex $v_{\alpha,\beta}$ for every local cluster $C_{\alpha,\beta}$, while the set $E$ contains the edge $e = (v_{\alpha_1,\beta_1}, v_{\alpha_2,\beta_2})$ only if their corresponding local clusters intersect, i.e., when $C_{\alpha_1,\beta_1} \cap C_{\alpha_2,\beta_2} \neq \emptyset$.

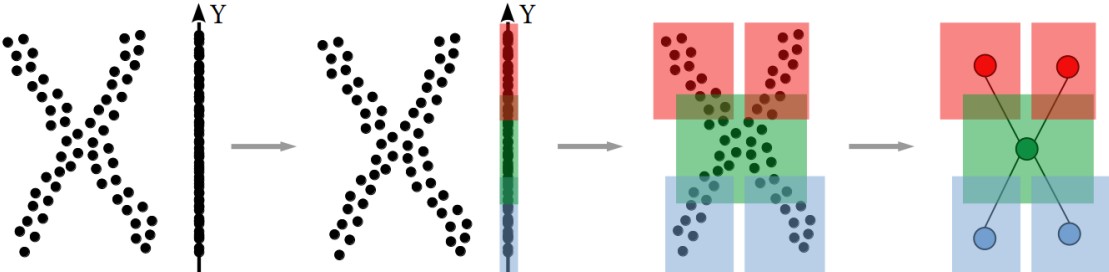

Figure 1: The four steps of Mapper on an X-shaped dataset where the lens is the projection on the $Y$-axis. Clusters from the same open set share the same color.

**The Nerve Theorem.** The theoretical foundation of Mapper is rooted in the *Nerve Theorem* (Borsuk, 1948; Weil, 1952), a fundamental result in algebraic topology that establishes a connection between a topological space and a combinatorial representation of its open covers. Specifically, it states that if an open cover $\mathcal{U}$ of a topological space $X$ forms a *good cover*, i.e. every finite intersection of sets in $\mathcal{U}$ is either empty or contractible, then the nerve $N(\mathcal{U})$, a simplicial complex encoding the intersections of sets in $\mathcal{U}$, is homotopy equivalent to $X$. The nerve $N(\mathcal{U})$ is constructed by associating a $(k-1)$-dimensional simplex to each non-empty intersection of $k$ sets in $\mathcal{U}$.

**Theorem 1.** *Let $X$ be a topological space and $\mathcal{U} = \{U_\alpha\}_\alpha$ be an open cover of $X$. If every finite intersection $U_{i_1} \cap \cdots \cap U_{i_k}$ is either empty or contractible, then $N(\mathcal{U})$ is homotopy equivalent to $X$.*

The Nerve Theorem is central to tools like Mapper because it enables the simplification of topological spaces into combinatorial structures, making them amenable to computational analysis. In the context of Mapper, the refined open cover $\mathcal{U}$ of $X$ (produced through clustering on the pullback cover) gives rise to the Mapper graph, which can be seen as the 1-dimensional truncation of $N(\mathcal{U})$. Here, 0-dimensional simplices correspond to nodes, and 1-dimensional simplices correspond to edges. When $\mathcal{U}$ is a good cover, the Mapper graph retains topologically relevant information about $X$. In particular:

- The connected components of the Mapper graph (0-cycles) accurately reflect the connected components of $X$.

- Loops in the Mapper graph (1-cycles) correspond to 1-cycles in $N(\mathcal{U})$, but not all such loops are topologically meaningful. Only 1-cycles that are *not boundaries* (i.e., elements of the 1-dimensional homology group $H_1(X)$) represent genuine features of $X$.

However, the Mapper graph cannot distinguish between true topological holes and 1-cycles that are boundaries (1-boundaries) in the full nerve $N(\mathcal{U})$, because information about higher-dimensional simplices (e.g., 2-simplices arising from triple intersections) is lost during truncation. Moreover, the clustering step may produce artifacts that violate the good cover condition, for example, by merging disconnected regions or disrupting simple-connectedness. In such cases, the correspondence between $X$ and the Mapper graph can break down. Nevertheless, when the good cover condition holds, the Mapper graph faithfully encodes the connected components and provides useful insights into the 1-dimensional topology of $X$.

The *good cover* condition explains why Mapper includes clustering in its steps. Clustering is used to split the open sets of the pullback, as a rough approximation of taking connected components. For example if the lens $f$ is projecting $X$ to a lower dimensional space, which is typical in the context of the classical Mapper, the clustering step becomes very important, since a projection can have multiple folds: if $B_p$ is a small ball at $f(p)$ the number of connected components of $f^{-1}(B_p)$ can jump when crossing critical values.

## 2.2 Standard Cubical Cover

In the original definition of Mapper (Singh et al., 2007), the authors use an open cover defined by two parameters: the *length* $w$ of the intervals and the *overlap* $p \in (0, 1/2]$, which is the fraction of $w$ that corresponds to the length $\delta$ of the intersection of any two adjacent intervals in the cover. This type of cover is sometimes referred to as a *cubical cover* or implicitly as a *standard cover* in the literature. In the rest of this work we will refer to such cover as *standard cubical cover*, and to any of its subcovers as *cubical cover*. We denote by $Y = f(X) \subseteq \mathbb{R}^k$ the space on which the cover is constructed.

**Definition 5.** Let $0 < n \in \mathbb{N}$ and $p \in (0, 1/2]$. Let $Y \subseteq \mathbb{R}$ compact. Let $m = min(Y)$, $M = max(Y)$, $w = \frac{M-m}{n(1-p)}$, $\delta = pw$ and define

$$a_j = m + j(w - \delta) - \delta/2, \quad b_j = m + (j+1)(w - \delta) + \delta/2.$$

The *standard cubical cover* of $Y$ with $n$ intervals and $p$ overlap is the collection of open sets

$$\mathcal{CC}_Y^{n,p} = \{(a_j, b_j) \cap Y_i | j = 0, \dots, n-1\} \, .$$

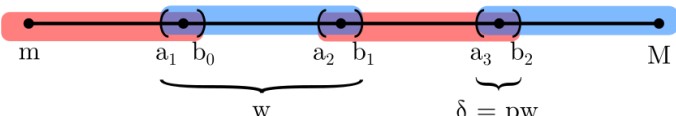

Figure 2: A visual representation of a standard cubical cover in the one-dimensional case with $n = 4$.

The value $w = \frac{M-m}{n(1-p)}$ correspond to the length of every interval, while $\delta = pw$ corresponds to the length of the overlap of any two adjacent intervals (see Figure 2).

**Definition 6.** Let $0 < n \in \mathbb{N}$ and $p \in (0, 1/2]$. Let $Y \subseteq \mathbb{R}^k$ compact and let $Y_i$ be the projection of $Y$ on the $i$-axis. The *standard cubical cover* of $Y$ with $n$ intervals and $p$ overlap is the collection of open sets

$$\mathcal{CC}_Y^{n,p} = \left\{ R \cap Y \neq \emptyset \middle| R \in \prod_{i=1}^{k} \mathcal{CC}_{Y_i}^{n,p} \right\} \, .$$

*Remark* 1. We report here some facts that easily follow from the definition in the case of a one-dimensional standard cubical cover for some $Y \subseteq [m, M]$.

- $b_i - a_i = w$ for every $i = 0, \dots, n-1$;

- $a_0 = m - \delta/2$ and $b_{n-1} = M + \delta/2$;

- $b_i - a_{i+1} = \delta$ for every $i = 0, \dots, n-1$.

- $[m, b_0 - \delta/2), \dots, [a_i + \delta/2, b_i - \delta/2), \dots [a_{n-1} + \delta/2, M]$ is a partition of $[m, M]$.

*Remark* 2. In this work we use a notion of standard cubical cover that simplifies some computations, but it's worth to point out that this definition may slightly diverge in literature. For example, some authors and software libraries define the standard cubical cover of $[n, M]$ in a way such that $a_0 = m$ and $b_{n-1} = M$. However, it's also important to remark that this definition is compatible with the results presented in Carrière et al. (2018).

**Naive Algorithm.** From Definition 6, we can readily devise an algorithm that initially computes the standard cubical cover for each projection independently. Subsequently, we assemble these individual open covers into an open cover for the topological space $Y$. We will refer to this approach as *naive construction* of standard cubical cover and is reported in Algorithm 2. This approach becomes computationally expensive for

high-dimensional datasets. This is also true in the case of Mapper, when we construct the standard cubical cover on high dimensional lenses. Even when many products are empty, their number can grow rapidly and introduce additional computational overhead to the entire Mapper process. To illustrate this issue, which is well known in literature, consider the following example: if $Y \subset \mathbb{R}^k$ lies along the diagonal, an appropriate cover for $Y$ could be achieved using a small number of rectangles, proportional to the number of intervals $n$. However, Algorithm 2 would construct an open cover for each projection initially and then iterate through all possible rectangles, resulting in a total of $n^k$ steps. As we will demonstrate later in Section 3, the primary contribution of this work is the resolution of this issue through the adoption of a more efficient algorithm. Instead of relying on projections, this algorithm iterates over a significantly smaller number of open sets, comparable to $n^{\dim(f(X))} \ll n^k$.

---

**Algorithm 2** Naive construction of the standard cubical cover

---

**Require:** $Y$ finite point cloud, $0 < n \in \mathbb{N}$, $p \in (0, 1/2]$.
**Ensure:** $\mathcal{CC}_Y^{n,p}$.
    **for** $i = 1, \ldots, k$ **do**
        $Y_i \leftarrow$ the projection of $Y$ on $i$-th axis;
        $\mathcal{CC}_{Y_i}^{n,p} \leftarrow \{I_{i,0}, \ldots, I_{i,n-1}\}$ the standard cubical cover on $Y_i$;
    **end for**
    $\mathcal{CC}_Y^{n,p} \leftarrow \{R = \prod_{i=1}^k I_{i,j_i} | R \neq \emptyset, 0 \leq j_i \leq n-1\}$.
    **return** $\mathcal{CC}_Y^{n,p}$

---

### 2.3 Ball Cover

When $Y \subseteq \mathbb{R}^k$, rectangles centered in $\mathbb{R}^k$ can form an open cover, as in the cubical covers. However, this approach is unsuitable for arbitrary metric spaces, where rectangles may not be well-defined. Instead, open balls centered at points in $Y$ offer a natural alternative.

In the context of Mapper, this idea has been explored in Dłotko (2019); Paweł Dłotko & Sazdanovic (2024), where the author introduces Ball Mapper, a variation of the original algorithm. In this approach, a greedy procedure is used to cover $Y$ with balls of fixed radius until full coverage is achieved. This method is sometimes referred to as an $\epsilon$-net, although the term also describes a related concept in computational geometry (Gonzalez, 1985) (see Definition 3). Notably, the centers derived in Ball Mapper satisfy the definition of an $\epsilon$-net according to Definition 3, making it straightforward to distinguish between these notions based on context.

**Definition 7.** Given a dataset $Y$ and a metric $d$ on $Y$, a *ball cover* of radius $r > 0$ on $Y$ is any open cover where every set is an open ball of radius $r$.

Unlike cubical covers, ball covers center their balls on points in $Y$, not evenly spaced points in $\mathbb{R}^k$. While balls in $L_\infty$-distance may appear cubical, the two constructions are fundamentally different.

**Cardinality of Ball Cover.** The cardinality of the ball cover determines the number of nodes in the Ball Mapper graph, directly influencing its complexity. While bounded above by $|Y|$, this bound is impractical for large datasets. Instead, $\epsilon$-nets provides a more useful estimate: the number of balls required is proportional to $(R/\epsilon)^{\dim(Y)}$, where $\dim(Y)$ is the doubling dimension of $Y$ (see Proposition 2).

### 2.4 Vantage Point Trees

Given a *query point $q$* and a *query radius $\epsilon$*, a *range query* is a function that returns the set of points within distance $\epsilon$ from $q$, i.e. the points in the ball $B(q, \epsilon)$. There are many ways to perform range queries efficiently, using different algorithms and data structures. A well-known example is the *kd-tree* (Friedman et al., 1977; Skrodzki, 2019), which partitions the space in a hierarchical tree-like structure that allows to reduce the number of distance computations employing the triangle inequality. The use of kd-trees in Mapper-type algorithms has been explored in Dłotko (2019), where the author notes that their effectiveness for Ball Mapper

may be limited, particularly in high-dimensional spaces where Ball Mapper typically operates. However, we believe that the approach presented in *Mapper on Ball Mapper* from Paweł Dłotko & Sazdanovic (2024) could benefit from incorporating a specialized data structure for range queries. In this case, the open cover is constructed on the space $f(X)$, which is often lower-dimensional than the original space $X$. Using a data structure optimized for range queries could thus offer a significant performance boost. In our study, we aim to address diverse scenarios by using any lens function $f : X \to Y$ with no restriction on the space $Y$. Importantly, $Y$ need not be strictly Euclidean or coordinate-based; it can encompass any domain where a meaningful notion of distance is defined. For all these reasons we decided to chose *vp-trees* instead of kd-trees (Yianilos, 1993; Brin, 1995).

*Remark* 3. It is important to note that vp-trees were our first choice due to their flexibility, as they can be used in any metric or pseudo-metric space. However, when $Y$ is a Euclidean domain contained within $\mathbb{R}^k$, it may be beneficial to explore other data structures that offer efficient spatial search. One such structure is R-trees (Guttman, 1984), which could be particularly well-suited for constructing cubical covers.

A *vantage-point tree*, or *vp-tree*, is a binary tree data structure where each internal node organizes the points of the space according to their distance from a chosen point, called *vantage point*. Each internal node stores a tuple $(p, r)$ as a reference to the ball $B(p, r)$ where $p$ is the chosen vantage point, and $p$'s descendants satisfy the *vp-tree property*: for every left descendant $y$ we have $d(p, y) \leq r$, and for every right descendant $z$ we have $d(p, z) \geq r$ (see also Figure 3).

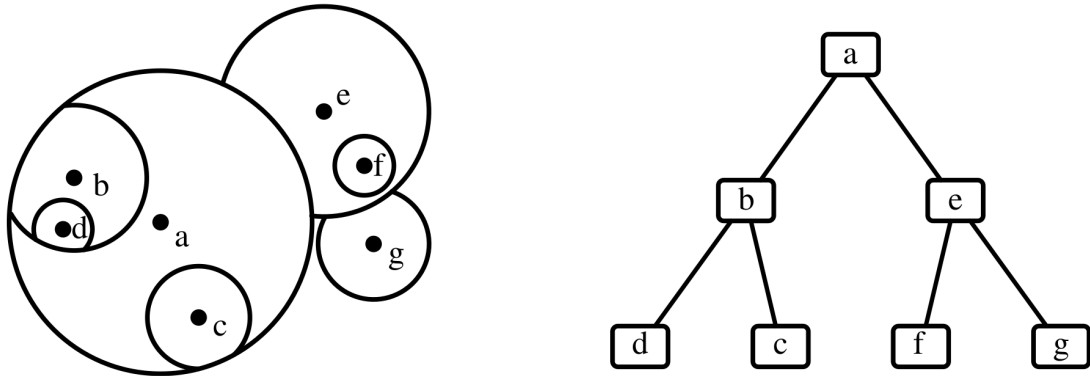

Figure 3: Two representations of a vp-tree on the dataset $Y = \{a, b, c, d, e, f, g\} \subseteq \mathbb{R}^2$. Nodes on the right correspond to balls on the left. The vp-tree property is evident: left descendants are enclosed by circles and right descendats are outside those circles.

**Building Vantage Point Trees.** The procedure used to build a vp-tree can be sketched in this way: given a dataset $Y$ we first chose a *vantage point $p$* from $Y$, then split $Y$ into two equally-sized subsets: those points that are closer to $p$, and those that are farther. Repeating the procedure on the two halves we obtain two trees $L$, obtained from the first half, and $R$ obtained from the second one. The result is then obtained as the binary tree rooted at $p$, with $L$ as left child and $R$ as right child. We will refer to this procedure as the *build_vptree* function which is reported in Algorithm 3

The time and space complexity of Algorithm 3 can be analyzed in the following way: given a dataset $Y$, every call to *build_vptree* can be implemented using an in-place procedure like quickselect on the input array $Y$, which takes $\mathcal{O}(|Y|)$ time and $\mathcal{O}(1)$ space. Therefore as a simple application of the master theorem, building a balanced vp-tree has asymptotic time complexity $\mathcal{O}(|Y| \cdot \log(|Y|))$ and asymptotic space complexity $\mathcal{O}(|Y|)$.

**Range Queries on Vantage Point Trees.** After a vp-tree is built, we can perform range queries by descending from the root (see Algorithm 4). Say we want to perform a range query for a point $q$ and radius $\epsilon$. Let $(p, r)$ be the tuple stored at any internal node while visiting the vp-tree. Using the triangle inequality it's possible to skip some of $p$'s children when certain conditions are met. In particular, we can do this in two situations: (a) when $B(q, \epsilon) \subseteq B(p, r)$, equivalent to $d(p, q) \geq r + \epsilon$, we need to visit only the left child

---

**Algorithm 3** Algorithm for $build\_vptree(Y, d)$

---

**Require:** Let $Y = [y_0, \ldots, y_{n-1}]$ be a dataset, and let $d$ be a metric on $Y$.
**Ensure:** $build\_vptree(Y, d)$ returns a vp-tree on $(Y, d)$.
  **if** $Y = \emptyset$ **then**:
    **return** $\emptyset$                                               ▷ the empty tree
  **else**
    $p \leftarrow$ choose in $Y$.                                        ▷ chose randomly
    Move $p$ at the head of $Y$, such that $y_0 = p$.
    Let $\rho = \text{median}_{y \in Y} d(p, Y)$.
    Reorder $Y$ such that $d(p, y_i) \leq \rho$ for $i < n/2$ and $d(p, y_i) \geq \rho$ for $i \geq n/2$.
    $L \leftarrow build\_vptree([y_1, \ldots, y_{n/2-1}], d)$
    $R \leftarrow build\_vptree([y_{n/2}, \ldots, y_{n-1}], d)$
    **return** $Tree\{root = (p, \rho), left = L, right = R\}$
  **end if**

---

(see Figure 4a); (b) when $B(q, \epsilon) \cap B(p, r) = \emptyset$, equivalent to $d(p, q) \geq r + \epsilon$, we need to visit only the right child (see Figure 4b).

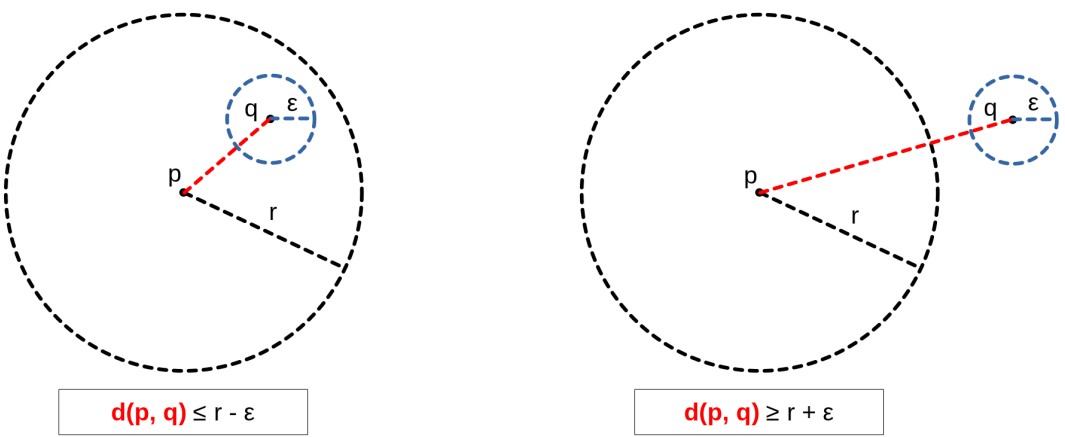

(a) When $B(q, \epsilon) \subseteq B(p, r)$ we skip the right child, since right descendants are outside the range query.

(b) When $B(q, \epsilon) \cap B(p, r) = \emptyset$ we skip the left child, since left descendants are outside the range query.

Figure 4: The two conditions when we need to visit only one child during range queries.

Range queries can be significantly more efficient with vp-trees than with linear scans. A linear scan requires going through all the points of a dataset $Y$, which takes $|Y|$ steps in total. On the other hand, with vp-trees, a range query usually takes less steps, since we can often skip one child from the search due to the triangle inequality satisfied by the metric (see Figure 4a and Figure 4b). Giving a general estimation of the average time complexity of range queries via vp-trees is particularly challenging due to its dependency on the dataset (Brin, 1995), we can only state that, for a dataset $Y$, it is bounded between $\mathcal{O}(\log(|Y|))$ and $\mathcal{O}(|Y|)$. However, when the query radius is sufficiently small, we expect to fall often in a cases where we can skip a branch from the range query. In such cases the time complexity becomes closer to $\mathcal{O}(\log(|Y|))$.

**Ball Cover via Vantage Point Trees.** It is worth emphasizing that the construction of the ball cover can be improved by leveraging vp-trees alone. Specifically, one can first construct a vp-tree $T$ on the dataset $Y$. Then, in the $\epsilon$-net algorithm, the open balls are generated using range queries on $T$ (see Algorithm 5).

---

**Algorithm 4** Algorithm for $range\_query(T, q, \epsilon)$

---

**Require:** Let $Y$ be a dataset, and $d$ a metric on $Y$. Let $T = build\_vptree(Y, d)$. Let $q$ be a query point, and let $\epsilon > 0$ be a query radius.
**Ensure:** The open ball $B(q, \epsilon)$.
  **if** $T$ is empty or terminal **then**
    **return** $\{y \in T.leaves \mid d(q, y) < \epsilon\}$
  **else**
    $(p, r) \leftarrow T.root$
    $S \leftarrow \emptyset$
    **if** $r < d(p, q) + \epsilon$ **then**                                              ▷ Opposite to Figure 4a
      $S \leftarrow S \cup range\_query(T.right, q, \epsilon)$
    **end if**
    **if** $r > d(p, q) - \epsilon$ **then**                                              ▷ Opposite to Figure 4b
      $S \leftarrow S \cup range\_query(T.left, q, \epsilon)$
    **end if**
    **return** $S$
  **end if**

---

**Algorithm 5** Construction of ball cover via vp-trees

---

**Require:** Let $(Y, d)$ be a metric space. Let $r \geq 0$.
**Ensure:** A ball cover $\mathcal{C}$ of radius $r$.
  $T \leftarrow build\_vptree(Y, d)$                                                        ▷ Algorithm 3
  $N \leftarrow \emptyset$
  $\mathcal{C} \leftarrow \emptyset$
  **while** $N \neq Y$ **do**
    Take $p \in Y \setminus N$
    $B \leftarrow range\_query(T, p, r)$                                              ▷ Algorithm 4
    $N \leftarrow N \cup B$
    $\mathcal{C} \leftarrow \mathcal{C} \cup \{B\}$
  **end while**
  **return** $\mathcal{C}$

---

## 3 Cubical Cover in Higher Dimensions

In this section, we outline the main contributions of this work. To begin, it is essential to introduce some notation.

**Definition 8.** Let $Y \subseteq \mathbb{R}^k$ compact. Let $m_i = \min_{y \in Y} y_i$, and $M_i = \max_{y \in Y} y_i$, with $m_i < M_i$ for $i = 1 \ldots k$. We define $\sigma_Y \colon \mathbb{R}^k \to \mathbb{R}^k$ by setting for every $y = (y_i)_{i=1,\ldots,k} \in \mathbb{R}^k$

$$\sigma(y) = \left( \frac{y_i - m_i}{M_i - m_i} \right)_{i=1\ldots,k}.$$

*Remark* 4. In the settings of Definition 8, the map $\sigma_Y \colon \mathbb{R}^k \to \mathbb{R}^k$ is a bijection that maps $Y$ to the hypercube $[0, 1]^k$. The map $\sigma^{-1} \colon \mathbb{R}^k \to \mathbb{R}^k$ is given by setting for each $y = (y_i)_{i=1,\ldots,k} \in \mathbb{R}^k$

$$\sigma_Y^{-1}(y) = (m_i + y_i(M_i - m_i))_{i=1,\ldots,k}.$$

**Definition 9.** Let $\rho_n \colon \mathbb{R}^k \to \left( \frac{1}{2n} + \frac{1}{n}\mathbb{Z} \right)^k$ be the map defined by setting for every $y = (y_i)_{i=1,\ldots,k} \in \mathbb{R}^k$

$$\rho_n(y) = \left( \frac{\lfloor ny_i \rfloor + 1/2}{n} \right)_{i=1,\ldots,k}.$$

Following Definition 6, we define a helper function that maps each point in $Y$ to its closest hypercube in the standard cubical cover. Specifically, this means the function assigns the hypercube whose center is the nearest neighbor to the point among all other hypercube centers.

**Definition 10.** Let $0 < n \in \mathbb{N}$ and let $p \in (0,1)$. Consider the interval $[m, M] \subseteq \mathbb{R}$ and let $w = \frac{M-m}{n(1-p)}$ and $\delta = pw$. Let $a_i = m + i(w - \delta) - \delta/2$ and $b_i = m + (i+1)(w - \delta) + \delta/2$. We define the *cubical proximity function* $CP_{[m,M]}(n, p) \colon [m, M] \to \mathcal{P}([m, M])$ by setting for every $y \in [m, M]$

$$CP_{[m,M]}(n, p)(y) = [m, M] \cap (a_i, b_i) \quad \forall y \in [a_i + \delta/2, b_i - \delta/2).$$

For any $Y \subseteq \mathbb{R}^k$ compact we can define for every $y = (y_i)_{i=1,\ldots,k} \in Y$

$$CP_Y(n, p)(y) = Y \cap \prod_{i=1}^{k} CP_{Y_i}(n, p)(y_i).$$

where $Y_i$ is the projection of $Y$ on the $i$-axis.

*Remark* 5. For every $y \in [m, M]$ there exists only one $j$ such that $y \in [a_j + \delta/2, b_j - \delta/2)$, which can be easily computed as

$$j = n \left\lfloor \frac{y - m}{M - m} \right\rfloor.$$

We can extend this to higher dimension. In case of $Y \subseteq \mathbb{R}^k$, for every $y = (y_i)_{i=1,\ldots,k} \in Y$, for each $i$ there exists only one $j_i = n \left\lfloor \frac{y_i - m_i}{M_i - m_i} \right\rfloor$ such that $y_i \in [a_{j_i} + \delta_i/2, b_{j_i} - \delta_i/2)$, and we have

$$CP_Y(n, p)(y) = Y \cap \prod_{i=1}^{k} (a_{j_i}, b_{j_i}).$$

This setting gives a well-defined notion for $CP_Y(n, p)$, since the intervals $[a_i + \delta/2, b_i - \delta/2)$ are a partition of $[m, M]$ (see Remark 1). We can finally state one of the main contributions of our methodology, in the form of the following result.

**Theorem 2.** *Let $Y \subseteq \mathbb{R}^k$ compact. Let $0 < n \in \mathbb{N}$ and let $p \in (0,1)$. Then for every $y \in Y$ we have*

$$CP_Y(n, p)(y) = B_{\sigma_Y^* d_\infty} \left( (\sigma_Y^{-1} \circ \rho_n \circ \sigma_Y)(y), \frac{1}{2n - 2np} \right).$$

*Proof.* We have $CP_Y(n, p)(y) = \prod_{i=1}^{k} (a_{j_i}, b_{j_i})$ where $j_i = n \left\lfloor \frac{y_i - m_i}{M_i - m_i} \right\rfloor$. Then

$$\sigma_Y(CP_Y(n, p)(y)) = \prod_{i=1}^{k} \left( \frac{a_{j_i} - m_i}{M_i - m_i}, \frac{b_{j_i} - m_i}{M_i - m_i} \right)$$

$$= \prod_{i=1}^{k} \left( \frac{j_i + 1/2}{n} - \frac{1}{2n - 2np}, \frac{j_i + 1/2}{n} + \frac{1}{2n - 2np} \right)$$

$$= B_{d_\infty} \left( \frac{j_i + 1/2}{n}, \frac{1}{2n - 2np} \right)$$

$$= B_{d_\infty} \left( \rho_n(\sigma_Y(y)), \frac{1}{2n - 2np} \right)$$

Therefore

$$CP_Y(n, p)(y) = \sigma_Y^{-1} B_{d_\infty} \left( \rho(\sigma_Y(y)), \frac{1}{2n - 2np} \right)$$

$$= B_{\sigma_Y^* d_\infty} \left( (\sigma_Y^{-1} \circ \rho \circ \sigma_Y)(y), \frac{1}{2n - 2np} \right)$$

$\square$

### 3.1 Estimating Cardinality

As a consequence of Theorem 2, we developed a more efficient method for constructing the elements of $\mathcal{CC}_Y^{n,p}$ using vp-trees. In this subsection, we provide an estimate of the cardinality of $\mathcal{CC}_Y^{n,p}$, which also allows us to assess the overall complexity of its construction. Theorem 3 establishes an upper bound on the cardinality of a minimal subcover of $\mathcal{CC}_Y^{n,p}$, while Corollary 1 extends this result to derive an upper bound on the cardinality of $\mathcal{CC}_Y^{n,p}$.

**Theorem 3.** *Let* $Y \subseteq \mathbb{R}^k$. *Then, there exist a subcover* $\mathcal{C} \subseteq \mathcal{CC}_Y^{n,p}$ *with cardinality*

$$|\mathcal{C}| \le \left(2n \cdot \frac{2-p}{p}\right)^{\dim(Y)}.$$

*Proof.* Initially we establish some notation that will make the proof easier. Let $\delta = \sigma_Y^* d_\infty$ and let $\psi_n = \sigma_Y^{-1} \circ \rho_n \circ \sigma_Y$. Under the metric $\delta$, $Y$ is contained in a $k$-dimensional hypercube of side 1, and $\psi_n$ acts as an approximation function that maps $Y$ to a regular grid of side $\epsilon_n = \frac{1}{2n}$. Then, as a consequence of Theorem 2, the collection $\mathcal{CC}_Y^{n,p}$ consists of the balls $B_\delta^Y(\psi_n(y), r_n)$ for each $y \in Y$, where the radius is $r_n = \frac{1}{2n-2np}$.

First, it's easy to observe that $\delta(y, \psi_n(y)) \le \epsilon_n$ for every $y \in Y$ and every $n$. Therefore every ball $B_\delta^Y(\psi_n(y), r_n)$ is contained within the ball $B_\delta^Y(y, r_n + \epsilon_n)$, which has the same center $y$ but a larger radius to account for the approximation error introduced by $\psi_n$. Therefore, for any chosen $m$, this gives us our first inclusion:

$$B_\delta^Y(\psi_m(y), r_m) \subseteq B_\delta^Y(y, r_m + \epsilon_m).$$

By recursively applying the notion of doubling dimension, the ball $B_\delta^Y(y, r_m + \epsilon_m)$ can be iteratively covered by $\lambda^s$ balls of radius $\frac{r_m + \epsilon_m}{2^s}$, where $s$ is the depth of the iteration and $\lambda$ is the doubling measure of $Y$ under the metric $\delta$. Therefore, we have:

$$B_\delta^Y(y, r_m + \epsilon_m) \subseteq \bigcup_{j=1}^{\lambda^s} B_\delta^Y\left(y_j, \frac{r_m + \epsilon_m}{2^s}\right),$$

where $\{y_j\}_j$ are the centers of the covering balls and $\lambda$ is the doubling measure of $Y$. If we now consider any $n \ge m$, using the same argument as in the first inclusion, we can write

$$B_\delta^Y\left(y_j, \frac{r_m + \epsilon_m}{2^s}\right) \subseteq B_\delta^Y\left(\psi_n(y_j), \frac{r_m + \epsilon_m}{2^s} + \epsilon_n\right),$$

which holds for any choice of $s$. If we then chose $s$ such that $\frac{r_m + \epsilon_m}{2^s} + \epsilon_n \le r_n$ we can further claim that

$$B_\delta^Y\left(\psi_n(y_j), \frac{r_m + \epsilon_m}{2^s} + \epsilon_n\right) \subseteq B_\delta^Y(\psi_n(y_j), r_n).$$

The inequality $\frac{r_m + \epsilon_m}{2^s} + \epsilon_n \le r_n$ can be easily solved in $s$, and gives $s \ge \log_2\left(\frac{n}{m} \cdot \frac{2-p}{p}\right)$, which holds when we set $s = \left\lceil \log_2\left(\frac{n}{m} \cdot \frac{2-p}{p}\right) \right\rceil$. After this, we can finally set $L = \lambda^s$ and give the following estimate:

$$L = \lambda^s = 2^{\dim(Y) \cdot s} \le 2^{\dim(Y) \cdot \left[1 + \log_2\left(\frac{n}{m} \cdot \frac{2-p}{p}\right)\right]} = \left(2 \cdot \frac{n}{m} \cdot \frac{2-p}{p}\right)^{\dim(Y)}.$$

Summing up and chaining the inclusions together, we obtain the following

$$B_\delta^Y(\psi_m(y), r_m) \subseteq \bigcup_{j=1}^{L} B_\delta^Y\left(y_j, \frac{r_m + \epsilon_m}{2^s}\right) \subseteq \bigcup_{j=1}^{L} B_\delta^Y(\psi_n(y_j), r_n).$$

Finally, setting $m = 1$ and $I_j = B_\delta^Y (\psi_n(y_j), r_n)$, we have $L \leq \left( 2n \cdot \frac{2-p}{p} \right)^{\dim(Y)}$ and

$$Y \subseteq B_\delta^Y (\psi_1(y), r_1) \subseteq \bigcup_{j=1}^{L} B_\delta^Y (\psi_n(y_j), r_n) = \bigcup_{j=1}^{L} I_j,$$

which concludes the proof. $\qquad\square$

**Corollary 1.** *Let $Y \subseteq \mathbb{R}^k$, then*

$$|\mathcal{CC}_Y^{n,p}| \leq 3^k \cdot \left( 2n \cdot \frac{2-p}{p} \right)^{\dim(Y)}.$$

*Proof.* Theorem 3 states that is always possible to find a subcover $\mathcal{S} \subseteq \mathcal{CC}_Y^{n,p}$ where $|\mathcal{S}| \leq \left( 2n \cdot \frac{2-p}{p} \right)^{\dim(Y)}$. Since $\mathcal{S}$ covers $Y$, every other interval $I \in \mathcal{CC}_Y^{n,p}$ must intersect some interval in $\mathcal{S}$. Therefore we can write

$$\mathcal{CC}_Y^{n,p} = \bigcup_{I \in \mathcal{S}} \mathcal{A}_I,$$

where $\mathcal{A}_I = \{ J \in \mathcal{CC}_Y^{n,p} \mid J \cap I \neq \emptyset \}$. It's easy to see that for dimensionality reasons $|\mathcal{A}_I| \leq 3^k$, therefore we can claim that

$$|\mathcal{CC}_Y^{n,p}| \leq |\mathcal{S}| \cdot 3^k \leq 3^k \cdot \left( 2n \cdot \frac{2-p}{p} \right)^{\dim(Y)},$$

and this concludes the proof. $\qquad\square$

Theorem 3 asserts that a minimal subcover of $\mathcal{CC}_Y^{n,p}$ has cardinality bounded by a value that depends solely on $n$, $p$, and $\dim(Y)$. This upper bound is *intrinsic* as it is independent from the dimension of the *feature space* $\mathbb{R}^k$. Conversely, the inequality in Corollary 1 is not intrinsic, as it also depends on $k$, yet it justifies why proximity-net runs in far fewer steps than $n^k$. Notably, this upper bound is a very rough estimation and could potentially be improved, as the factor $3^k$ is significantly higher than what is typically observed. While a smaller factor might be achievable, it remains unclear how such an improvement would be influenced by the specific dataset.

**Standard Cubical Cover via Vantage Point Trees.** Theorem 2 suggests how we can construct the hypercubes of the standard cubical cover as open balls under a scaled $L_\infty$-distance. This insight leads to an immediate improvement in constructing the standard cubical cover: first, a vp-tree $T$ is built using the scaled $L_\infty$-distance. Then, after identifying all the hypercubes and their centers (noting that some centers may not correspond to points in the dataset), the points within each hypercube can be efficiently retrieved using range queries on $T$ centered at these points (see Algorithm 6).

While this improvement is significant, it is still insufficient. Although it reduces the number of steps compared to Algorithm 2, a single point in the dataset may still lie in the intersection of up to $2^k$ open hypercubes. This detail is often overlooked but has critical implications: the standard cubical cover could, in principle, contain more open sets than there are points in the dataset. This issue becomes particularly pronounced in higher dimensions, where such open covers tend to produce Mapper graphs that are too complex to provide meaningful insights.

### 3.2 Proximity-Net

In this work, we introduce a generalization of $\epsilon$-net that we call *proximity-net*. This modified algorithm is a greedy procedure that takes a single parameter, that we call *proximity function* (see Definition 11), and covers the dataset with a collection of sets.

**Definition 11.** A *proximity function* on $Y$ is a map $b \colon Y \to \mathcal{P}(Y)$ such that $p \in b(p)$ for any $p \in Y$.

---

**Algorithm 6** Construction of the standard cubical cover via vp-trees

---

**Require:** Let $Y \subseteq \mathbb{R}^k$, let $0 < n \in \mathbb{N}$ and $p \in (0, 1/2]$.
**Ensure:** The standard cubical cover $\mathcal{CC}_Y^{n,p}$.
  $T \leftarrow build\_vptree(Y, \sigma_Y^* d_\infty)$                                              $\triangleright$ Algorithm 3
  $L \leftarrow (\sigma_Y^{-1} \circ \rho_n \circ \sigma_Y)(Y)$
  $\mathcal{C} \leftarrow \left\{ range\_query \left( T, l, \frac{1}{2n-2np} \right) \ \middle| \ l \in L \right\}$                $\triangleright$ Theorem 2, Algorithm 4
  **return** $\mathcal{C}$

---

*Remark* 6. The cubical proximity function $CP_Y(n, p)$ from Definition 10 is a proximity function according to Definition 11.

The proximity-net algorithm is reported in Algorithm 7 and the only difference with respect to $\epsilon$-net is that the sets obtained from proximity-net are built by applying the proximity function, and therefore are not required to be open balls. This choice brings improved flexibility and allows to build diverse types of open covers by applying the same procedure to a properly chosen parameter. In this section we will see how we can obtain both the ball cover and a cubical cover using proximity-net. More importantly, deriving a cubical cover from proximity-net effectively addresses the flaw of Algorithm 2, as the number of open balls is expected to be significantly fewer than $n^k$ (Corollary 1).

---

**Algorithm 7** Construction of proximity-net

---

**Require:** Let $Y$ be a dataset, and let $b$ be a proximity function on $Y$.
**Ensure:** A cover of $Y$
  $S \leftarrow Y$, as a set                         $\triangleright$ $S$ is the set that tracks the points of $Y$ not covered yet
  $C \leftarrow \emptyset$
  **while** $S \neq \emptyset$ **do**
    Take a point $p \in S$                              $\triangleright$ Randomly or according to some heuristic
    $B \leftarrow b(p)$
    Add $B$ to $C$
    **for** $q \in B$ **do**                             $\triangleright$ All the points in $B$ are now covered
      Remove $q$ from $S$
    **end for**
  **end while**
  **return** $C$

---

It's worth to point out that the original $\epsilon$-net can be obtained by supplying proximity-net with the *ball proximity function* defined as in Definition 12, and further optimize it using vp-trees. This optimization is reported in Algorithm 8 and is essentially the same as Algorithm 5.

**Definition 12.** Let $Y \subseteq Y'$ and let $d$ be a pseudo-metric on $Y'$. For each $\epsilon > 0$ we define the function $BP_Y(d, \epsilon) \colon Y' \to \mathcal{P}(Y)$ by setting

$$BP_Y(d, \epsilon) \colon y \mapsto Y \cap B_d(y, \epsilon).$$

for every $y \in Y'$. Moreover, the restriction of $BP_Y(d, \epsilon)$ on $Y$ is a proximity function that we call *ball proximity function*.

*Remark* 7. Definition 12 allows to use the same notation $BP_Y$ when we want to construct a ball with a center that is not contained in $Y$, but in an eventually larger space $Y'$. We will use this in Theorem 2.

We can improve $\epsilon$-net algorithm by first building a vp-tree $T$ on the dataset to be covered. After that we can call proximity-net (Algorithm 7) by supplying a function that for each point $p$ performs a range query on $T$. This approach (Algorithm 8) is eventually faster than the original $\epsilon$-net approach.

*Remark* 8. The ability to work with pseudo-metrics, rather than just metrics, is an invaluable feature of vp-trees that we can leverage in our implementation. In the setting of Mapper on Ball Mapper, we have a lens $f \colon X \to Y$ and a metric $d$ on $Y$. Mapper on Ball Mapper is obtained by taking the pullback of the open sets of Ball Mapper under the lens $f$. This is equivalent to apply Algorithm 8 to the input dataset $Y = X$

---

**Algorithm 8** Construction of $\epsilon$-net via proximity-net and vp-trees

---

**Require:** Let $Y$ be a dataset and $d$ a pseudo-metric on $Y$. Let $\epsilon > 0$ be a chosen radius.
**Ensure:** A ball cover $\mathcal{C}$ on $Y$ with balls of radius $\epsilon$.

   $T \leftarrow build\_vptree(Y, d)$                                        ▷ Algorithm 3
   $\pi \leftarrow p \mapsto range\_query(T, p, \epsilon)$                        ▷ Definition 12, Algorithm 4
   $\mathcal{C} \leftarrow proximity\text{-}net(X, \pi)$                                ▷ Algorithm 7
   **return** $\mathcal{C}$

---

under the pullback pseudo-metric $f^*d$. This brings a practical benefit in terms of time and space, since the pullback cover is already obtained in this way, without constructing it explicitly from a cover on $f(X)$.

**Cubical Cover via Proximity-Net and Vantage Point Trees.** Under an appropriate choice of proximity function, we can construct a cubical cover using proximity-net, while keeping the number of open sets limited, as in the case of $\epsilon$-net (see Remark 2.3), eliminating the performance degradation encountered in Algorithm 2.

*Remark* 9. As a result of Theorem 2, we can claim that

$$y \mapsto B_{\sigma_Y^* d_\infty}\left((\sigma_Y^{-1} \circ \rho_n \circ \sigma_Y)(y), \frac{1}{2n - 2np}\right)$$

is a proximity function.

By leveraging Theorem 2 we can finally summarize our methodology for computing this cover efficiently, which is also reported in Algorithm 9. In the first step we use Algorithm 3 to construct a vp-tree $T$ on $Y$ using the pseudo-metric $\sigma_Y^* d_\infty$. The range query method on $T$ (Algorithm 4) is then equivalent to computing the proximity function $BP_Y(\sigma_Y^* d_\infty, \epsilon)$ for any choice of $\epsilon$. Then, once $T$ has been constructed, we run proximity-net algorithm (Algorithm 7) by supplying the proximity function $BP_Y\left(\sigma_Y^* d_\infty, \frac{1}{2n-2np}\right)(\sigma_Y^{-1} \circ \rho_n \circ \sigma_Y)$ which by definition is equivalent to

$$y \mapsto B_{\sigma_Y^* d_\infty}\left((\sigma_Y^{-1} \circ \rho_n \circ \sigma_Y)(y), \frac{1}{2 - 2p}\right),$$

and therefore can be efficiently computed as

$$y \mapsto range\_query\left(T, (\sigma_Y^{-1} \circ \rho_n \circ \sigma_Y)(y), \frac{1}{2n - 2np}\right)$$

using the vp-tree $T$. As stated by Theorem 2, this is equivalent to computing $CP_Y(n, p)(y)$.

---

**Algorithm 9** Construction of cubical cover via proximity-net and vp-trees

---

**Require:** Let $Y \subseteq \mathbb{R}^k$, let $0 < n \in \mathbb{N}$ and $p \in (0, 1/2]$.
**Ensure:** A cubical cover $\mathcal{C} \subseteq \mathcal{CC}_Y^{n,p}$.

   $T \leftarrow build\_vptree(Y, \sigma_Y^* d_\infty)$                                ▷ Algorithm 3
   $\pi = y \mapsto range\_query(T, (\sigma_Y^{-1} \rho_n \sigma_Y)(y), \frac{1}{2n-2np})$    ▷ Definition 10, Theorem 2, Algorithm 4
   $\mathcal{C} \leftarrow proximity\text{-}net(\pi)$                                   ▷ Algorithm 7
   **return** $\mathcal{C}$

---

Since proximity-net is a greedy algorithm that selects a distinct element of $\mathcal{CC}_Y^{n,p}$ at each step, estimating the cardinality of $\mathcal{CC}_Y^{n,p}$ also provides an upper bound on the total number of iterations of proximity-net. Consequently, the algorithm produces an open cover of $Y$, where each open set corresponds to one of the hyperrectangles from the original standard cubical cover $\mathcal{CC}_Y^{n,p}$. This refined open cover may contain fewer open sets than the original cubical cover, thanks to the application of proximity-net. Despite its smaller size, the cover remains sufficient to encompass the entire dataset. Moreover, the Mapper graph derived from this open cover retains its informativeness while being potentially easier to visualize and analyze, particularly in higher dimensions (see Figure 5).

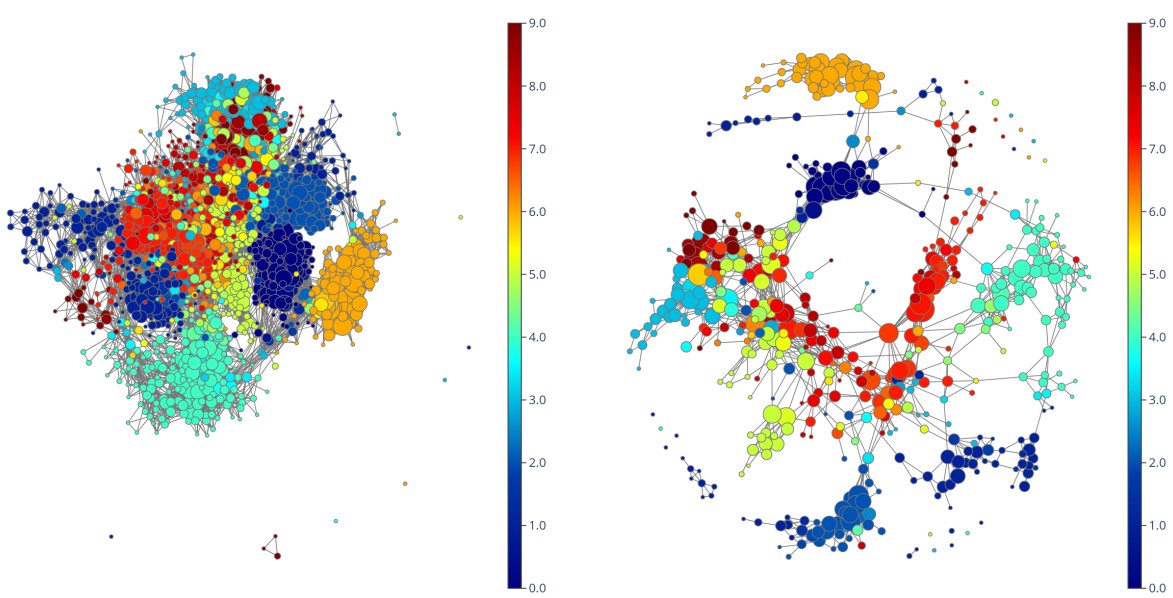

Figure 5: A visual comparison between the standard cubical cover (left) and the cover from proximity-net (right) on the *Digits* dataset (Alpaydin & Kaynak, 1998). Node colors represent the average digit values. We ran Mapper using PCA with four principal components as the lens, ten intervals and 50% overlap, and KMeans clustering with two clusters.

**Non-Determinism.** Algorithm 7 does not prescribe a strict rule for selecting points in its main loop, reflecting the inherent non-determinism of $\epsilon$-net construction as defined in Algorithm 1. This flexibility means that the resulting Mapper graph may vary depending on the specific choice of points.

On one hand, it's possible to reduce this variability by picking, at each step, the point $p$ that maximizes the distance $d(p, Y \setminus S)$. While this approach increases computational cost, it can improve consistency and reproducibility.

On the other hand, when the *good cover* condition is met, the impact of non-determinism on the Mapper graph is not topologically relevant, since as discussed more deeply in 2.1, the topological information encoded in the Mapper graph does not depend on the specific open cover. If this condition is not met, the Mapper graph may no longer reflect the topological features of $X$, and using Mapper in such cases is questionable since the Nerve Theorem does not apply.

Moreover, when the open cover derived from proximity-net is a subcover of a good cover, it inherently satisfies the good cover condition. For instance, in the case of a standard cubical cover—constructed deterministically—if the good cover condition is satisfied for the standard cubical cover, then the standard Mapper graph is topologically meaningful, and the Mapper graph obtained via proximity-net also preserves the same topological features. Under these conditions, the choice of points in Algorithm 7 does not affect the topological features of the Mapper graph.

**Computational Complexity.** The computational complexity of Algorithm 7 can be estimated if we know the cardinality of the open cover produced via the $\epsilon$-net or proximity-net and the complexity of the proximity function. This estimation is particularly relevant for two cases: the ball proximity function $BP_Y(d, \epsilon)$ and the cubical cover proximity function $CP_Y(n, p)$. In both cases, range queries are performed using vp-trees, corresponding to Algorithm 8 for the ball cover and Algorithm 9 for the cubical cover.

For Algorithm 8, the main loop iterations correspond to the size of the ball cover, which is $\mathcal{O}((R/r)^{\dim(Y)})$, as stated in Proposition 2. However, the time complexity of range queries, that we denote by $\Psi_Y(r)$, is challenging to estimate, as also reported by Uhlmann (1991). Additionally, constructing the vp-tree on a

dataset $Y$ has an asymptotic time cost of $\mathcal{O}(|Y| \cdot \log(|Y|))$ and takes $\mathcal{O}(|Y|)$ additional space. Combining these factors, we can write:

$$Time(Y, r) = \mathcal{O}\left(|Y| \cdot \log(|Y|) + \Psi_Y(r) \cdot \left(\frac{R}{r}\right)^{\dim(Y)}\right)$$

$$Space(Y, r) = \mathcal{O}\left(|Y| + \left(\frac{R}{r}\right)^{\dim(Y)}\right).$$

For the cubical cover, and $Y \subseteq \mathbb{R}^k$, we estimate the cardinality of the open cover using Theorem 2. The process is analogous, leading to:

$$Time(Y, n, p) = \mathcal{O}\left(|Y| \cdot \log(|Y|) + \Psi_Y\left(\frac{1}{2n - 2np}\right) \cdot 3^k \cdot \left(2n \cdot \frac{2-p}{p}\right)^{\dim(Y)}\right)$$

$$Space(Y, n, p) = \mathcal{O}\left(|Y| + 3^k \cdot \left(2n \cdot \frac{2-p}{p}\right)^{\dim(Y)}\right).$$

### 3.3 Experimental Results

To evaluate the benefits of the approach outlined in Algorithm 8, and supported by Theorem 3 and Corollary 1, we conducted a series of programmatic experiments. Initially, we developed a Python library called *tda-mapper* (Simi, 2024) based on the approach presented in this work. Subsequently, we compared it against other open-source libraries. The motivation behind creating *tda-mapper* was to explore alternative methods for constructing open covers for Mapper and eventually implement a more efficient approach. While major open-source implementations like *Python Mapper (v0.1.17)* (Müllner & Babu, 2013), *GUDHI (v3.10.1)* (Carrière, 2024), *giotto-tda (v0.6.0)* (Tauzin et al., 2021), and *Kepler Mapper (v2.0.1)* (van Veen et al., 2019) can in principle support high-dimensional lenses, they all rely on Algorithm 2 which has known limitations, as previously discussed. The root cause of this issue lies in their source code: a common thread among these libraries is the use of the `itertools.product` function. This function, described in Python's official documentation available at `https://docs.python.org/3/library/itertools.html#itertools.product`, is used to perform a nested loop on each one-dimensional open cover, which corresponds to what Algorithm 2 does.

In this section, we report the results obtained from comparing *giotto-tda (v0.6.2)* (Tauzin et al., 2021), *Kepler Mapper (v2.1.0)* (van Veen et al., 2019) and *tda-mapper (v0.9.0)* (Simi, 2024), focusing on both running time performance and the complexity of the generated graphs. These results align with the expected behavior and demonstrate the clear superiority of our approach, which achieves better scalability with respect to lens dimension while producing less complex graphs. To evaluate the performance and scalability of our approach, we conducted a series of measurements on the running time required to compute Mapper graphs. During these benchmarks, we consistently kept a fixed number of intervals and overlap, while systematically varying the lens dimension. This comparative analysis provides valuable insights into the behavior of these implementations when dealing with high-dimensional data. Our experiments were conducted on Debian 12 using Python 3.11, using a PC equipped with a Ryzen 7 5700G CPU with 2x16GB DDR34 2133Mhz, in dual channel configuration. To ensure the reliability of our benchmarks, we used well-known datasets publicly available at the UCI Machine Learning Repository (Dua & Graff, 2023): (a) the *Digits* dataset (Alpaydin & Kaynak, 1998), with 1797 instances and 64 features; (b) the *MNIST* dataset (LeCun et al., 1998), with 70000 instances and 784 features; (c) the *Cifar-10* dataset (Krizhevsky et al., 2009), with 60000 instances and 1024 features; and (d) the *Fashion-MNIST* dataset (Xiao et al., 2017) with 70000 instances and 784 features. For each dataset we ran Mapper using overlap fraction $p$ ranging in the set $\{0.125, 0.25, 0.5\}$ and using $n = 10$ as the number of intervals on each feature. This choice is arbitrary, but was enough to get informative Mapper graphs, especially with low-dimensional lenses, with every dataset we used.

As a final note, it is important to emphasize that our experiments were largely constrained by memory limitations. Many instances could not be executed for values of $k > 5$ due to out-of-memory errors encountered while benchmarking *giotto-tda* and *kepler-mapper*. Consequently, direct comparisons are limited to $k \leq 5$. Nonetheless, the asymptotic behavior of *tda-mapper* has been analyzed for higher values of $k$, up to $k = 10$. These benchmarks highlight the good scalability properties of our approach in this extended range.

**Choosing the open cover.** In the following plots we report the running times of *tda-mapper* on the cubical cover via proximity-net (Algorithm 7) and on the ball cover via $\epsilon$-net (Algorithm 8).

It's important to point out that these two open covers don't align and need different input parameters, so comparing the two approaches requires a little care. For this reason the ball cover has been constructed by supplying the inputs that better match those of cubical cover, i.e. as the *metric* we chose the scaled $L_\infty$-distance, and as the *radius* we chose $1/(2n - 2np)$ (see Proposition 2).

As we will see in the plots there is often no clear winner in terms of performance between cubical cover and ball cover, except for a single case where the ball cover scales better with dimension than the cubical cover.

**Choosing the clustering algorithm.** The choice of clustering algorithm plays a crucial role in the complexity of the Mapper graph. Different clustering approaches may result in varying numbers of nodes and edges, impacting both the interpretability of the graph and its computational cost. For example, clustering algorithms that require the number of clusters $k$ as an input parameter (such as KMeans (MacQueen, 1967)) would produce a Mapper graph with $s\,k$ nodes, where $s$ is the number of open sets constructed on the image of the lens. Certain clustering algorithms are particularly well-suited to specific lenses. For instance, when using density as the lens, DBSCAN (Ester et al., 1996) might be a good choice. This is because the open sets in the pullback cover will often correspond to regions of approximately uniform density, aligning well with DBSCAN's focus on density.

However, in the context of constructing Mapper graphs using high-dimensional lenses (one of the key aspects of our method) the reliance on clustering diminishes. High-dimensional lenses often yield open sets that can naturally distinguish clusters without requiring additional clustering (see the discussion in 2.1 about the Nerve Theorem). This principle can also be observed in Dłotko (2019), where the open cover is directly constructed on the dataset rather than in the lens image, eliminating the need for clustering altogether. Similarly, in our approach, the higher-dimensional lens spaces reduce the dependency on clustering compared to classical Mapper, where lenses are typically low-dimensional (e.g., 2D projections).

Given these considerations, and because our experiments focus on the asymptotic behavior when the lens dimension is high, we chose a trivial clustering strategy. Specifically, we opted for a clustering algorithm that assigns all data points to a single cluster. This minimizes the influence of clustering on our benchmarks and isolates the effects of our approach for building open covers in our analysis.

**Scaling with the Embedding Dimension.** The first experiment involves creating a 1-dimensional dataset embedded in dimension $k$, referred to as the *Line* dataset in the plots. This is a toy experiment where the dataset consists of 10000 points lying on the diagonal of the hypercube $[0, 1]^k$, with a small random noise. As expected, compared with *kepler-mapper* and *giotto-tda*, the running time of *tda-mapper* on this dataset demonstrates the advantage of our approach especially in higher dimensions (see Figure 6).

**Scaling with the Intrinsic Dimension.** We conducted additional experiments to better reflect typical use cases when employing Mapper libraries (see Figure 7, 8, 9, 10). To streamline the process, we used *Principal Component Analysis* (PCA) as the lens, varying the number of components from 1 to 10. As the number of PCA components $k$ increases, the discrepancy with the doubling dimension of the image is also expected to grow. This range was sufficient to highlight a significant performance advantage of *tda-mapper* in all experiments for $k \geq 4$. In contrast, both *kepler-mapper* and *giotto-tda* encountered frequent out-of-memory issues, and their results are shown only for experiments that successfully ran. Memory consumption poses a major challenge to algorithm scalability, and notably, *tda-mapper* completed all experiments without any out-of-memory errors, consistently using far less memory. The plots depict running times on a linear scale for the main axes, supplemented by a logarithmic scale in the inset plots. Interestingly, as $k$ increases,

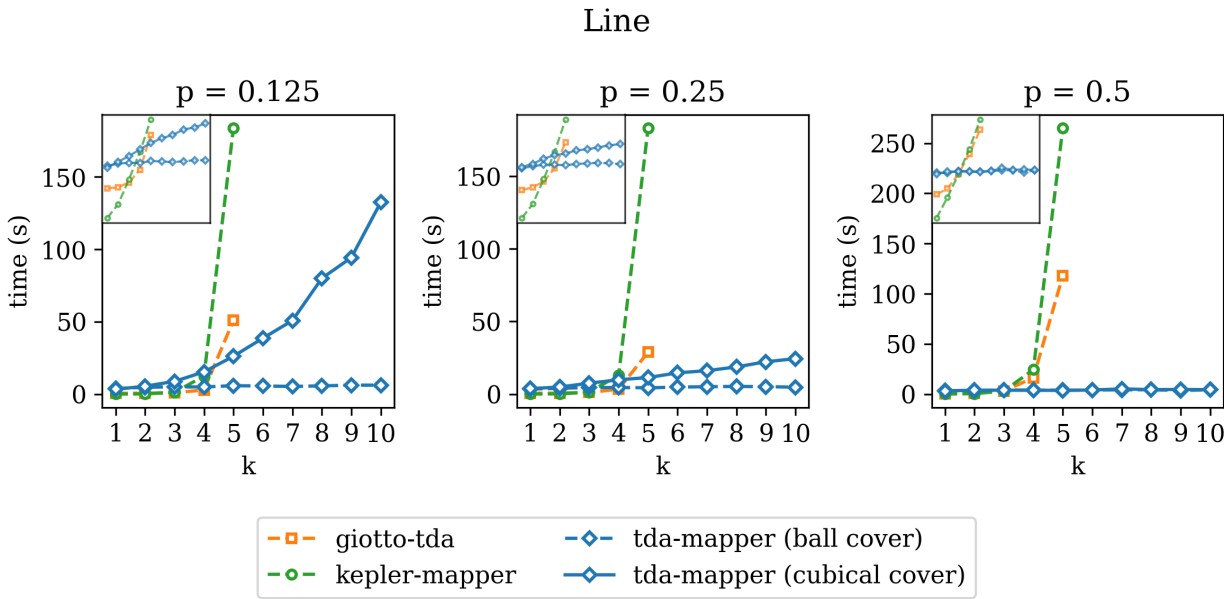

Figure 6: Comparison of running times on the *Line* dataset in dimension *k*.

*tda-mapper* exhibits sub-exponential growth in running time. Extending these experiments to larger *k* values would be worthwhile to determine whether running times stabilize (as observed in the *Digits* dataset) or continue to grow.

### 3.3.1   Visualization

To highlight the visual improvements achieved with proximity-net, we present comparisons of Mapper graphs generated from our benchmark datasets in Figures 11, 13, 15, and 17. The Mapper algorithm was configured using PCA with four principal components as the lens, a cubical cover with five intervals and 50% overlap, and KMeans clustering with two clusters was applied to the pullback of each open set. Nodes are colored based on their labeled classes. As expected, the Mapper graphs generated by *giotto-tda* and *kepler-mapper* are less intuitive to navigate compared to those produced by *tda-mapper*. However, the simpler graphs from *tda-mapper* effectively capture relationships among the different color-coded classes, offering clearer insights compared to the other libraries.

Additionally, we compare the reported graphs across five distinct metrics in all experiments: *density*, *transitivity*, *degree*, *clustering coefficients*, and *betweenness centrality*. Detailed comparisons of the node-level metrics (degree, clustering coefficients, and betweenness centrality) are presented in Figures 12, 14, 16, and 18 as histograms, showing their distributions across all nodes. Tables 1, 2, 3, and 4 provide aggregated values for these distributions, reported as means and standard deviations (std), along with the graph-level metrics density and transitivity. Notably, transitivity and clustering coefficients exhibit similar patterns across the three libraries tested. This suggests that all libraries maintain similar levels of local clustering, preserving the tendency for nodes within a neighborhood to form tightly-knit clusters. However, we observe several differences in other metrics. Compared to *giotto-tda* and *kepler-mapper*, *tda-mapper* exhibits lower average degrees, yet maintains the same clustering coefficients and transitivity. Despite having fewer connections overall, the graphs generated by *tda-mapper* preserve similar levels of local cohesiveness, suggesting a focus on more concentrated, direct connections within clusters. A key distinction arises in betweenness centrality, where *tda-mapper* consistently demonstrates higher values. This indicates that certain nodes play a more pivotal role in bridging different parts of the graph, pointing to a more centralized or hierarchical structure. These nodes are crucial for connecting otherwise isolated clusters. Furthermore, the higher density observed in *tda-mapper* suggests that its graphs are more compact, with fewer nodes but denser interconnections, potentially offering a clearer and more streamlined representation of the key relationships between clusters.

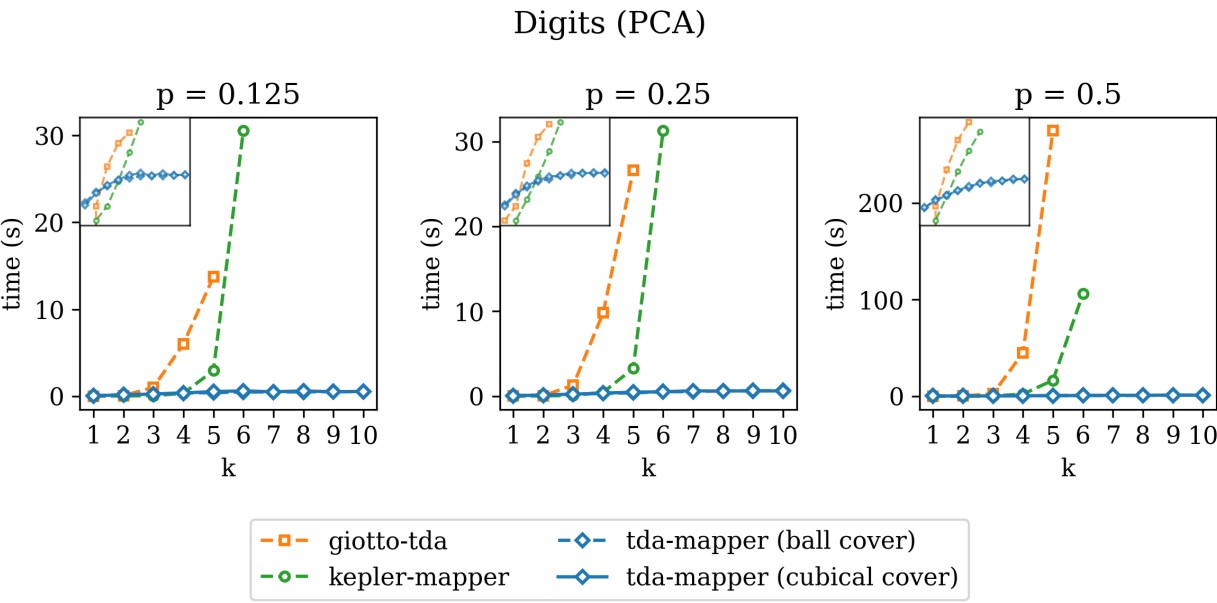

Figure 7: Comparison of running times on the PCA with $k$ components on the *Digits* dataset.

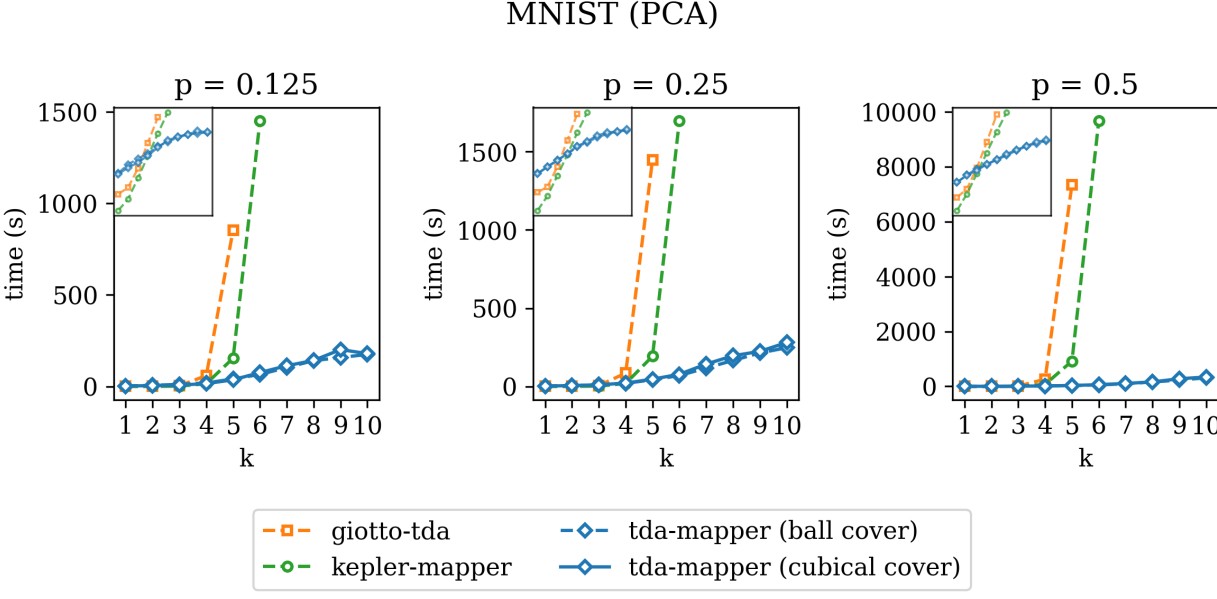

Figure 8: Comparison of running times on the PCA with $k$ components on the *MNIST* dataset.

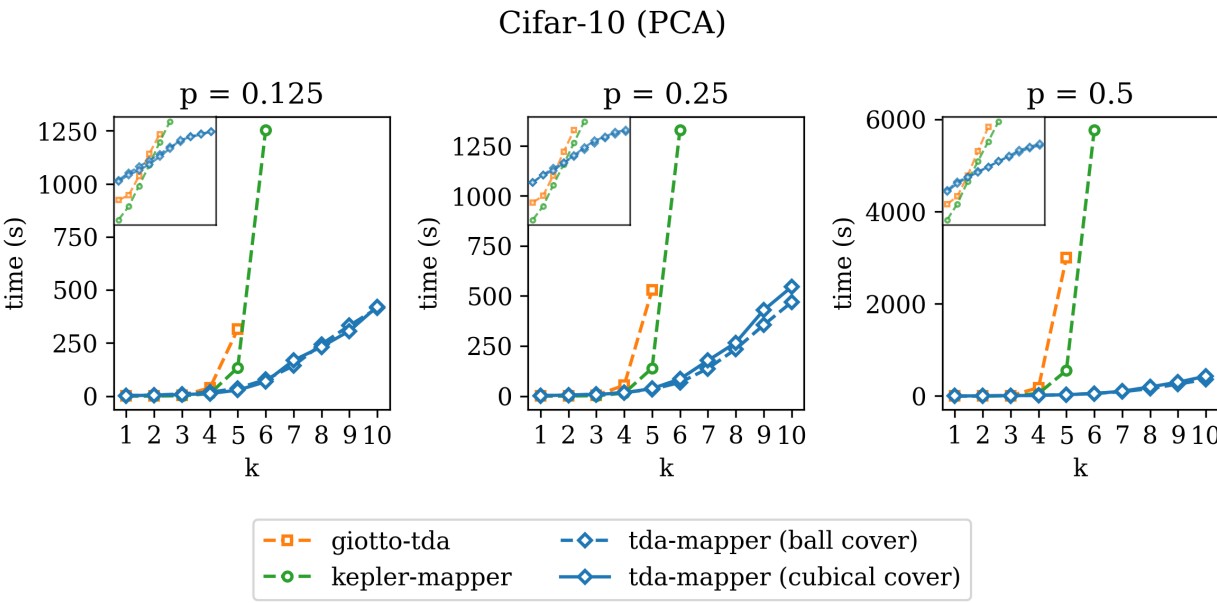

Figure 9: Comparison of running times on the PCA with $k$ components on the *Cifar-10* dataset.

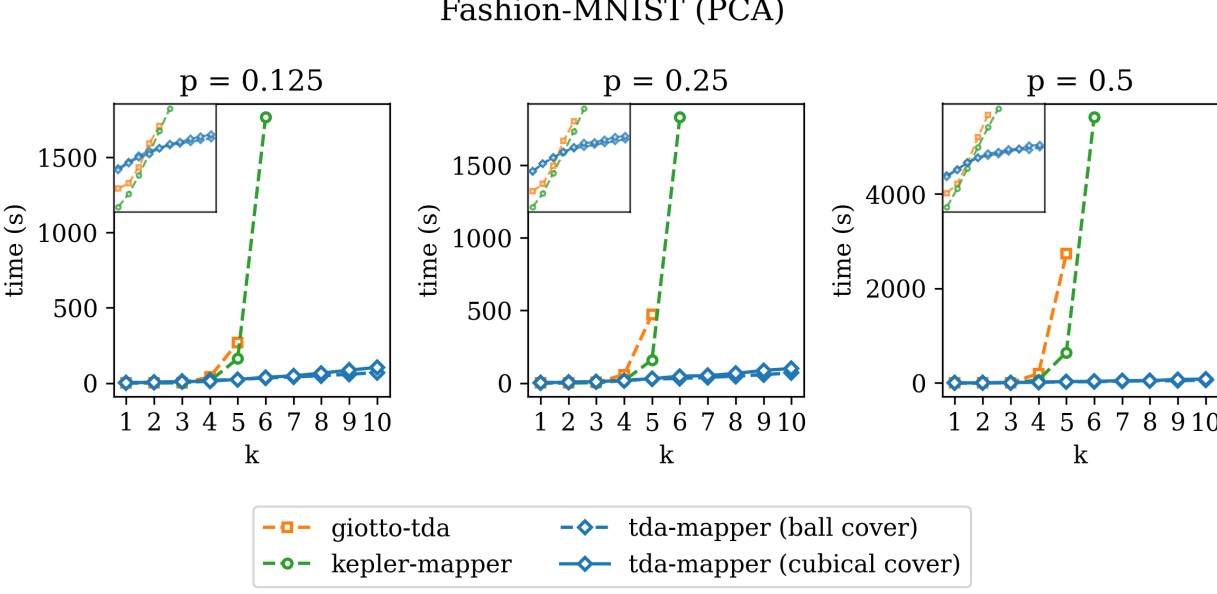

Figure 10: Comparison of running times on the PCA with $k$ components on the *Fashion-MNIST* dataset.

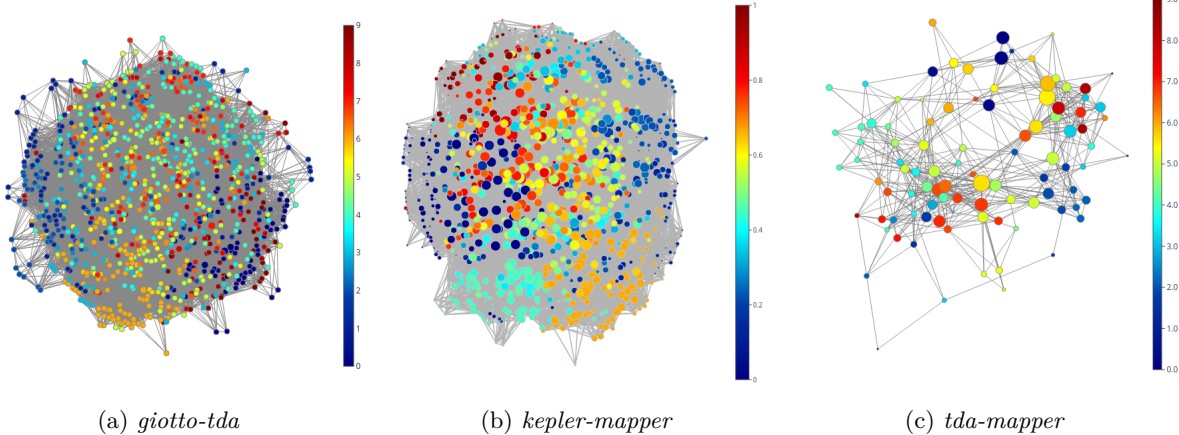

(a) *giotto-tda*                    (b) *kepler-mapper*                    (c) *tda-mapper*

Figure 11: Comparison of Mapper graphs for *Digits*.

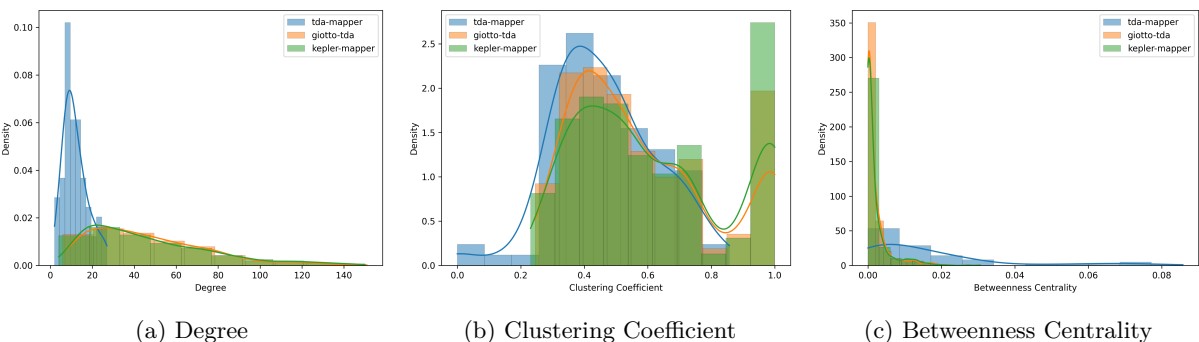

(a) Degree                    (b) Clustering Coefficient                    (c) Betweenness Centrality

Figure 12: Comparison of metrics for *Digits*.

Table 1: Summary metrics for *Digits*.

| Metric | giotto-tda | kepler-mapper | tda-mapper |
|---|---|---|---|
| Density | 0.04 | 0.05 | 0.12 |
| Transitivity | 0.40 | 0.40 | 0.41 |
| Degree (std) | 46.27 (27.51) | 46.82 (29.64) | 11.80 (5.77) |
| Clustering (std) | 0.58 (0.22) | 0.61 (0.24) | 0.46 (0.16) |
| Betweenness (std) | 0.002 (0.003) | 0.002 (0.004) | 0.016 (0.019) |

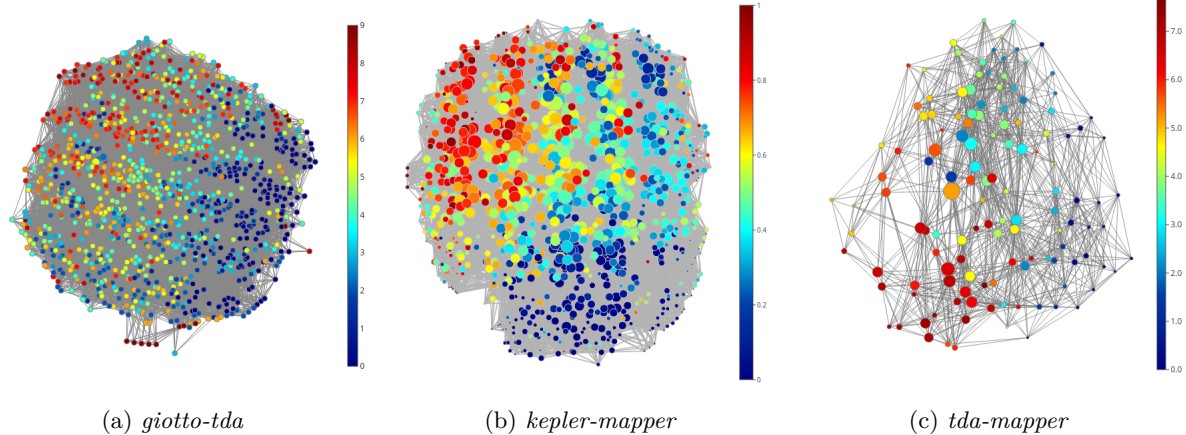

(a) *giotto-tda*          (b) *kepler-mapper*          (c) *tda-mapper*

Figure 13: Comparison of Mapper graphs for *MNIST*.

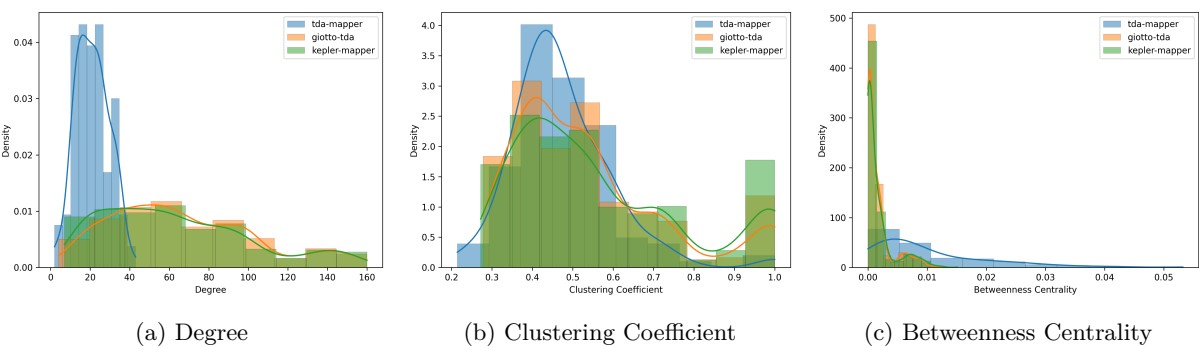

(a) Degree          (b) Clustering Coefficient          (c) Betweenness Centrality

Figure 14: Comparison of metrics for *MNIST*.

Table 2: Summary metrics for *MNIST*.

| Metric | giotto-tda | kepler-mapper | tda-mapper |
|---|---|---|---|
| Density | 0.06 | 0.06 | 0.16 |
| Transitivity | 0.40 | 0.40 | 0.44 |
| Degree (std) | 66.78 (36.50) | 63.58 (37.62) | 21.15 (8.29) |
| Clustering (std) | 0.53 (0.19) | 0.56 (0.21) | 0.47 (0.12) |
| Betweenness (std) | 0.002 (0.002) | 0.002 (0.003) | 0.01 (0.01) |

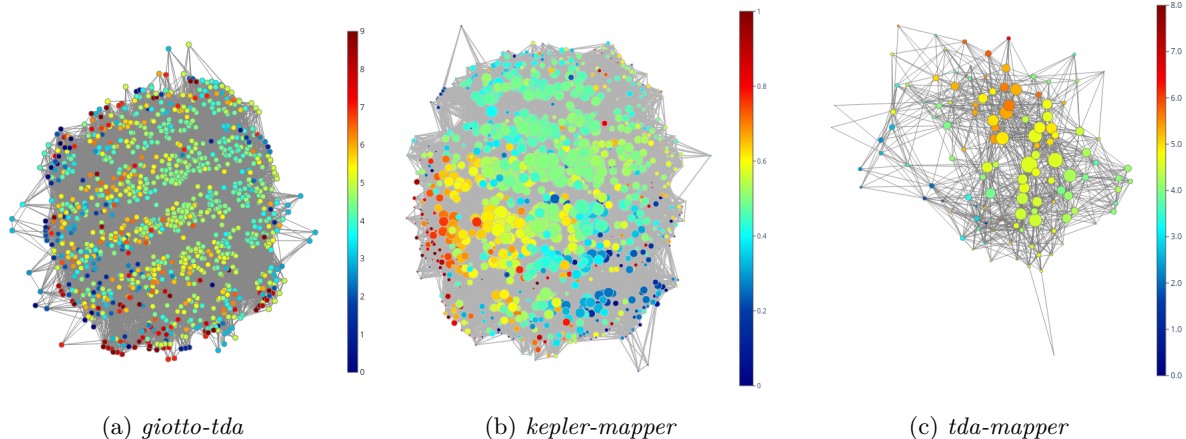

(a) *giotto-tda*  (b) *kepler-mapper*  (c) *tda-mapper*

Figure 15: Comparison of Mapper graphs for *Cifar-10*.

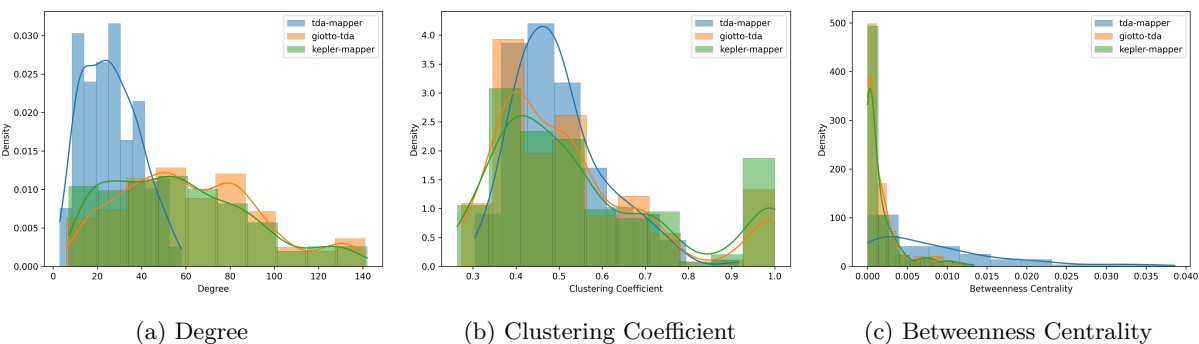

(a) Degree  (b) Clustering Coefficient  (c) Betweenness Centrality

Figure 16: Comparison of metrics for *Cifar-10*.

Table 3: Summary metrics for *Cifar-10*.

| Metric | giotto-tda | kepler-mapper | tda-mapper |
|---|---|---|---|
| Density | 0.05 | 0.05 | 0.18 |
| Transitivity | 0.40 | 0.40 | 0.45 |
| Degree (std) | 62.25 (31.23) | 57.54 (31.73) | 26.08 (12.43) |
| Clustering (std) | 0.53 (0.19) | 0.56 (0.21) | 0.50 (0.11) |
| Betweenness (std) | 0.002 (0.002) | 0.002 (0.003) | 0.008 (0.008) |

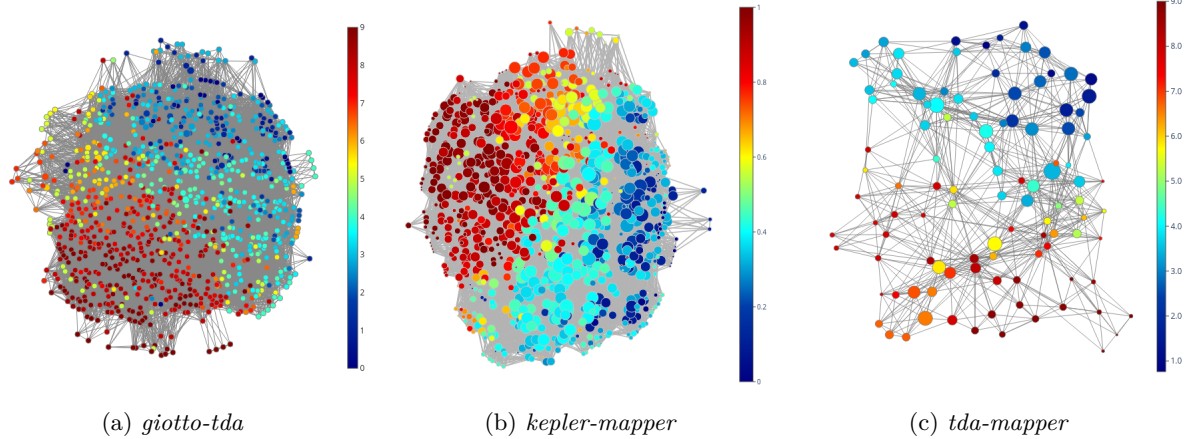

(a) *giotto-tda*      (b) *kepler-mapper*      (c) *tda-mapper*

Figure 17: Comparison of Mapper graphs for *Fashion-MNIST*.

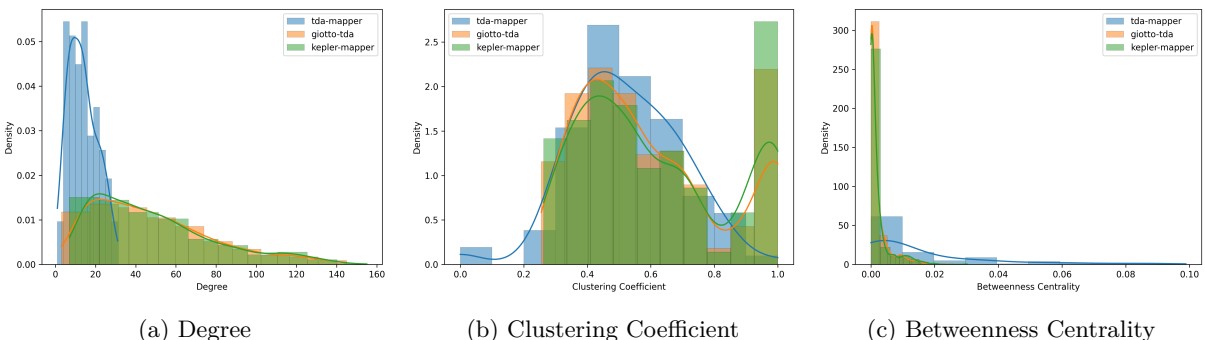

(a) Degree      (b) Clustering Coefficient      (c) Betweenness Centrality

Figure 18: Comparison of metrics for *Fashion-MNIST*.

Table 4: Summary metrics for *Fashion-MNIST*.

| Metric | giotto-tda | kepler-mapper | tda-mapper |
|---|---|---|---|
| Density | 0.05 | 0.05 | 0.13 |
| Transitivity | 0.40 | 0.40 | 0.43 |
| Degree (std) | 48.09 (30.15) | 48.46 (30.36) | 13.58 (7.06) |
| Clustering (std) | 0.59 (0.23) | 0.61 (0.23) | 0.52 (0.17) |
| Betweenness (std) | 0.002 (0.003) | 0.002 (0.004) | 0.015 (0.021) |

**Acknowledgements**

This work is dedicated to the memory of my father, whose support and encouragement have been a constant source of strength throughout my life. I wish to express my deep gratitude to Sofia Torchia for her insightful comments and genuine interest in this work, which have been invaluable. I also extend sincere thanks to the anonymous reviewers for their constructive feedback and suggestions, which have greatly improved this paper, and to the action editor for their commitment to the review process.

This research was conducted independently and did not receive any external funding or institutional support. The findings presented here are not associated with the author's professional duties or affiliations.

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

# A  Appendix: Library Overview

Throughout the development process of *tda-mapper*, one of the objectives was to create an API that is easy to understand and use. For this reason we adopted an object-oriented approach taking inspiration from the well-known *scikit-learn* APIs (Pedregosa et al., 2011), since we expect some good level of familiarity with it from the intended user base of *tda-mapper*. Additionally, we made efforts to keep the API of *tda-mapper* similar to the APIs provided by *giotto-tda* and *Kepler Mapper*, allowing users to smoothly transition between these libraries and leverage their existing knowledge. By considering these factors, we aim to provide a user-friendly and seamless experience for users of *tda-mapper*, making it easier for them to explore and use the library's full potential.

We have implemented our own version of the vp-tree data structure and optimized it for our specific use-case: our implementation allows each leaf of the vp-tree to contain multiple items by stopping the construction when the splitting circle is *small*, either in terms of its cardinality or in terms of its radius (smaller than a given threshold). This optimization is beneficial both for range queries and for K-nearest neighbor (KNN) queries. When, during a search, the visited node becomes smaller than the query, the search operation collapses into a faster brute force linear scan.

The implementation of *tda-mapper* relies on several dependencies, including `networkx` (Hagberg et al., 2008), `numpy` (Harris et al., 2020), `matplotlib` (Hunter, 2007), and `plotly` (Plotly Technologies Inc., 2015). Overall, the software dependencies in *tda-mapper* are crucial for its functionality and enable users to generate Mapper graphs and visualize them effectively:

- `networkx` is used to generate and manipulate the Mapper graph, which is the primary result of the algorithm.

- `numpy` is necessary for numeric computations, particularly for the `CubicalCover` function.

- `matplotlib` and `plotly` are used to create plots for the Mapper graph, providing visualization options.

Additionally, there is a weaker dependency on `sklearn` (Pedregosa et al., 2011) which is used only for testing, ensuring that the implementation aligns with widely-used machine learning standards. The `sklearn` library is used to check that the custom-defined estimators in *tda-mapper* are compatible with `sklearn`. An extensive amount of effort was devoted to ensure a good level of automation during development, especially for testing, which is performed using GitHub actions. At the time of writing code coverage is around 96%.

For more in depth information, examples, tutorials, and documentation, the interested reader can visit `https://tda-mapper.readthedocs.io/en/main/`.

