# OpenReview forum: "A Scalable Approach for Mapper via Efficient Spatial Search"
_TMLR — Accepted by TMLR_

### Review · Reviewer_AQbC · 2024-12-01

**Summary Of Contributions:**

This paper introduces a  approximate but scalable approach for constructing open covers in the Mapper algorithm, a key tool in Topological Data Analysis (TDA). The contributions of the paper are outlined below:
(1)) The paper introduces proximity-net, a generalization of the \epsilon-net algorithm, to construct open covers more efficiently.
(2) The approach uses vantage-point trees (vp-trees), a specialized data structure, to efficiently handle range queries and reduce the computational complexity of building open covers, especially for high-dimensional datasets.
(3) The authors provide theoretical upper bounds on the cardinality of the constructed covers, showing that the number of open sets depends on the dataset's intrinsic dimensionality and the resolution parameters, rather than growing exponentially with the feature space dimensionality. Therefore, this serves as a complexity analysis, along with the scaling experiments.

**Audience:**

Yes

**Broader Impact Concerns:**

I would thank the author for contributing to the important research topic of making TDA/mapper algorithms scalable. Its applications on existing machine learning dataset and comparison against the state-of-the-art libraries makes a convincing case for using better implementation of mapper with proximity-net for ML applications.
The paper could attract even more audience if it chooses TDA venue like SoCG or DCG.

**Claims And Evidence:**

Yes

**Requested Changes:**

(1) The construction proximity-net is quite similar to the chaining technique in probability (c.f., the classic Ledoux & Talagrand. Probability in Banach Spaces: isoperimetry and processes, 2013.) where the covering is defined by semi-metrics induced by covariance operators.
The authors are applying this to reduce the cardinality of covering sets, which is natural from a algorithmic standpoint. The idea of counting the cardinality of covering balls to approximate actual dimensionality of the targeted manifold has also been developed from an asymptotic angle within the TDA community, to name a few:
Adler, Robert J., Omer Bobrowski, and Shmuel Weinberger. "Crackle: The homology of noise.", 2014.
Luo, Hengrui, Steven N. MacEachern, and Mario Peruggia. "Asymptotics of lower dimensional zero-density regions.", 2023.
It would be helpful to see how this line of work in asymptotics can be connected to the motivation of this scalable method.

(2) Since the author introduced a new concept of dimensionality called doubling dimension, it would be helpful to see how this new concept conincide with traditional concept of dimensionality (e.g., in a Hilbert vector space, does this doubling dimension coincide with the usual dimension?), to establish an intuition for the reader. If some example spaces are given, could we use this concept to estimate the computational complexity of the proposed algorithm?

(3) According to the literature, I would suggest that the authors to revise the beginning of section 2 to include how previous work used the \epsilon-net in the context of TDA, and perhaps drop some notions (e.g., Def 3 L_\infty distance can be assumed to be known or put into appendices) if we assumed that the audience has some prequisites.
From my expertise, I think the novelty of the current paper is to concretely construct VP-tree to obtain an efficient search regime without running into the memory limit brought by the search tree structure, then the authors should put emphasis on this contribution in their writing.

(4) For Algorithms 7 and 8, can the author present a classic big O complexity analysis? Currently, the complexity analysis is briefly discussed only in remarks (e.g., Rem 5)

(5) The writing style is clean but can be more concise, for example on page 17 "This dual-scale approach..." explains the choice of plotting, which is not too relevant to the main message. For another example is the remark 8, which should occur once the choice of vp-tree is made.

(6) When the author presented scaling results like Figures 6-8, is there a convenient way to allow the reader to know how much performance loss by adopting a proximity-net instead of the standard \epsilon-net? If such a comparison is not possible, could we have an explanation or guidance about what is "lost" when using the proximity-net instead of the \epsilon-net?

(7) Continuing (6) and the second point in weakness, while the proposed method demonstrates superior performance compared to PCA,  but not to widely used dimensionality reduction techniques such as t-SNE and UMAP. A comparative analysis with these modern methods would significantly enhance the paper, showcasing its strengths relative to popular alternatives and increasing its appeal to a broader audience.

(8)The clustering step is briefly described, with the authors opting for a trivial algorithm that produces a single cluster per input dataset to minimize its impact on benchmarks. However, the influence of clustering on the resulting Mapper graph’s complexity and interpretability warrants deeper exploration. For instance, investigating the method’s sensitivity to different clustering algorithms could provide valuable insights into its robustness and practical utility.

Points (2) and (6) are critical; other points, especially (1) and (7) can greatly strengthen the paper.

**Strengths And Weaknesses:**

Strengths:
(1)The paper presents theoretical guarantees, including bounds on the cardinality of covers, ensuring the scalability of the approach. A strong mathematical foundation ties the new method to established results in computational geometry and TDA. I think this will give the audience more confidence even without complexity analysis.

(2) Experimental results on diverse datasets highlight the practicality and efficiency of the proposed approach.

Weakness:
(1) Although the method performs better than PCA, the practical upper limits of the approach for high-dimensional datasets (e.g., >1,000 features) are not thoroughly analyzed or discussed. In practice, t-SNE and UMAP are more widely used and showed successful results, even on some of the datasets chosen by the authors (e.g., CIFAR-10), but not compared against with. I think comparison with modern dimension reduction method will enhannce the paper quite a bit and attract audience's interest.

(2) The clustering step is briefly mentioned (i.e., "To minimize the effect of clustering on the benchmarks, we chose a trivial clustering algorithm that for each input dataset creates a single cluster."), but its role in graph complexity and interpretability could be explored in more detail. For instance, how sensitive is the method to different clustering algorithms?

---

> ### Author Response · Authors · 2024-12-20
>
> We appreciate the reviewer's valuable suggestions and will integrate them into the next revision of the manuscript. We also thank the reviewer for helping us clarify the points outlined in the review. These contributions will surely strengthen this work. Below, we address the points one by one:
>
> # Weaknesses
>
> 1. **Comparing Mapper with PCA, t-SNE, and UMAP**
>
>     While all serve data exploration, comparisons are context-dependent and vary by application. Mapper is not strictly a dimensionality reduction technique, so direct comparisons with PCA, t-SNE, or UMAP are challenging. PCA was used as a lens in Mapper rather than a competitor. We will clarify this in the revision and consider benchmarks using UMAP as a lens.
>
> 2. **Clustering Sensitivity**
>
>     Clustering impacts Mapper's graph complexity and interpretability. However, high-dimensional lenses, a focus of our method, reduce reliance on clustering by constructing open sets that naturally distinguish clusters. This principle aligns with "Ball Mapper" by P. Dlotko, which constructs open covers without clustering. For the same reason, the importance of clustering in our approach is often less relevant than in the context of the classical Mapper, where the lens is typically a 2-dimensional projection. We will emphasize this distinction and include a discussion on clustering sensitivity and its implications.
>
> # Requested Changes
>
> 1. **Incorporating References**
>
>     We will review the suggested references and discuss how our approach aligns with or differs from these prior works.
>
> 2. **Doubling Dimension**
>
>     We will clarify that the doubling dimension is an established concept, referencing works like Gupta, Krauthgamer, and Lee's "Bounded geometries, fractals, and low-distortion embeddings", as well as Assouad's "Plongements lipschitziens dans $\mathbb{R}^n$". To establish intuition, we will include the following key result (which is contained in the previous references) in the revised text: in Euclidean spaces $\mathbb{R}^k$ under any $l_p$-metric, the doubling dimension is proportional to the ambient dimension $k$, i.e., $dim(\mathbb{R}^k) = \mathcal{O}(k)$. This result ensures that the doubling dimension aligns with our geometric intuition in classical vector spaces.
>
> 3. **Role of $\epsilon$-nets**
>
>     We will expand on $\epsilon$-nets' applications in TDA, referencing works like "Ball Mapper" and "Mapper-type algorithms." Greater emphasis will be placed on the VP-tree construction, highlighting its importance and addressing space complexity concerns, particularly for high dimensions.
>
> 4. **Complexity Analysis**
>
>     Big-O complexity analyses for Algorithms 7 and 8 will be added, integrated with existing discussions in Remark 5, and tied to $\epsilon$-net construction and cubical cover cardinality.
>
> 5. **Remark Placement**
>
>     Remark 8 will be relocated after the VP-tree discussion, with phrasing revised for clarity and relevance.
>
> 6. **Proximity-net vs. $\epsilon$-net**
>
>     Unlike a standard $\epsilon$-net, the proximity-net ensures that the resulting cover is a subcover of the cubical cover. This property is crucial for preserving the compatibility with results obtained in scientific literature for Mapper, for example a classical result is the possibility to estimate optimal parameters for Mapper, as show in "Statistical Analysis and Parameter Selection for Mapper" by M. Carrière, B. Michel, S. Oudot. Regarding performance comparisons, in a direct application of $\epsilon$-net to Mapper we must remark that $\epsilon$-net does not align with the cubical cover. This misalignment means that $\epsilon$-net cannot directly replace proximity-net. Nonetheless, the $\epsilon$-net approach is a valuable construction, and the idea of comparing the scalability and efficiency of proximity-net with a ball-based cover derived from $\epsilon$-net is interesting and will be added in the revised manuscript.
>
> 7. **Mapper vs. t-SNE/UMAP**
>
>     Mapper, t-SNE, and UMAP differ fundamentally: Mapper summarizes data topology, while t-SNE and UMAP focus on geometric embeddings in low dimensions. Considered these differences, comparisons in terms of scalability and runtime could be done, but they could also mislead the reader into thinking that Mapper is an alternative to those. Regarding qualitative comparisons, the distinct goals of Mapper and t-SNE/UMAP make it challenging to evaluate their outputs on the same metrics. While t-SNE and UMAP focus on the geometric arrangement of data points in a low-dimensional embedding, Mapper emphasizes the topology of the data, summarizing its shape and connectivity.
>
> 8. **Clustering Algorithms**
>
>     The clustering algorithm choice affects the Mapper graph, but is less critical in case of high-dimensional lenses. While an exhaustive analysis of all clustering methods is beyond the scope of this work, we will highlight specific cases where different clustering algorithms might influence Mapper.

---

> > ### Comment · Reviewer_AQbC · 2024-12-27
> >
> > Thanks for the efforts in revision. I am ok with the revised version.
> >
> > The author put more emphasis in using nerve theorem for better illustration for the role of clustering, but did not show how the dimensionality affects the choice of lens; I want to point out that in higher dimensional spaces, it is not trivial to select a robust clustering algorithm for Mapper, especially on datasets that contains certain sparsity like networks (outside of the standard datasets used in the paper).
> >
> > Reviewer qUwr critically pointed out the uncertainty associated with the algorithm, which questions the algorithmic robustness when the dataset contains outliers in high dimensions. I did not feel this point well discussed in the revision.

---

> > > ### Author Response · Authors · 2024-12-28
> > >
> > > We thank reviewer AQbC for their thoughtful comments and for pointing out important considerations. We appreciate the opportunity to address these points and provide additional clarification.
> > >
> > > Our algorithm modifies the standard Mapper pipeline by generating a subcover of the full standard cubical cover. After clustering, this results in a subgraph of the classical Mapper graph. Consequently, our approach does not introduce additional artifacts to the output. Instead, it offers two key advantages: (a) faster output generation and (b) a more interpretable graph. The challenges raised by the reviewer are inherent to Mapper-type algorithms in general and are not specific to our method, nor do we claim to resolve them. Specifically:
> > >
> > > ## 1. Clustering and dimensionality
> > >
> > > We agree that the choice of a robust clustering algorithm is critical to the Mapper pipeline. However, the sensitivity to clustering is a well-documented challenge for all Mapper-based methods, not unique to our approach. Our method focuses on the construction of the open cover and makes no assumptions about the clustering step. The stability of clustering, particularly for 1-dimensional lenses $f: X \to \mathbb{R}$, has been extensively studied in works such as:
> > >
> > > - "Statistical Analysis and Parameter Selection for Mapper" (M. Carrière, B. Michel, S. Oudot)
> > > - "Structure and Stability of the 1-Dimensional Mapper" (M. Carrière, S. Oudot)
> > > - "A Numerical Measure of the Instability of Mapper-Type Algorithms" (F. Belchí, J. Brodzki, M. Burfitt, M. Niranjan).
> > >
> > > We do not claim to tackle these issues (it would be out of scope for our paper) or to diminish their importance in any way.
> > >
> > > Moreover, we point out that clustering is always applied in the dataset space, not in lens space. Therefore, choosing a high-dimensional lens does not affect the dimensionality of the clustering task.
> > >
> > > We also point out that it would be ideal for Mapper to choose a lens which matches the dimension of the original dataset, as it could retain more information than a low-dimensional one. This option has always been out of the table because it was practically unfeasible to use lenses with dimension > 3 (and this is confirmed in the benchmark plots of this paper).
> > >
> > > In some cases the clustering step can be bypassed. For example (with the limitations of the theoretical point of view) if $X$ is a smooth manifold and the lens $f$ is smooth with no critical point (ideally, when $f$ is the identity on $X$), the level sets of $f$ are single points. When $X$ is covered with sufficiently small balls, each ball would naturally contain a single connected component, eliminating the need for clustering. In such cases fine-tuning the open cover could be enough.
> > >
> > > ## 2. Outliers
> > >
> > > Outlier detection in high dimensions is a very general problem, inherently complex, and we make no claims to address it. However, the non-deterministic nature of our algorithm does not affect this issue. In general, outliers may be statistically irrelevant, but not topologically irrelevant. The algorithm that we call proximity-net in the paper is very similar to the epsilon-net from "Clustering to minimize the maximum intercluster distance" (T. F. Gonzalez) and similarly to that, fully covers the dataset (in this case the image of the lens) without excluding any points, including outliers. From Mapper's perspective, outliers typically manifest as small or sparse connected components in the resulting graph. This behavior is consistent across both the standard Mapper and our proposed methodology. As such, our methodology does not introduce additional challenges regarding outliers.

---

### Review · Reviewer_qUwr · 2024-12-16

**Summary Of Contributions:**

This paper presents an efficient method to compute cubical covers for Mapper graphs. First, the paper identifies a combinatorial problem in existing methods to compute cubical covers for Mapper graphs, which leads to long running times and inflated number of nodes in Mapper graphs. Then, the paper introduces a series of optimizations, such as using vp-trees and efficient range queries. Finally, the paper introduces $\epsilon$-proximity net as an efficient way of obtaining a cubical cover for the given dataset. The proposed method is benchmarked for speed against two existing libraries that allow computing cubical covers, namely `giotto-tda`and `kepler-mapper`, showing improved speed in higher dimensions.

**Audience:**

Yes

**Broader Impact Concerns:**

No concerns.

**Claims And Evidence:**

Yes

**Requested Changes:**

* Could the Mapper graphs for Fashion-MNIST, MNIST, and CIFAR (or representative subsets of these datasets) be added to the paper? This would help the reader to get a better impression of the advantages of the propsed method in comparison to existing libraries.
* Could a discussion be added up to which dimension the proposed method is still feasible, or is there really only an advantage with $k$=5 (In Fig. 6 to 10)?
* Could a comparison of Mapper Graphs with different random seeds be shown, to assess the impact of non-determinism in Algorithm 7?

In summary, I think the paper is already very clear and makes an interesting contribution, but especially the non-determinism of Algorithm 7 should be addressed, as it could have important implications regarding the validity of resulting graphs.

**Typos**
  * Page 1: "A well-knonw"
  * Page 1: "Singh et al. Singh et al. (2007).": Double citation?
  * Page 2: "These crucial points has been" -> "have been"

**Strengths And Weaknesses:**

**Strengths**
  * Very detailed and clear introduction to all relevant concepts and methods
  * Practical and useful contribution to the community
  * Method is released as a Python package

**Weaknesses**\
While I think the main part of the paper is technically sound and easy to follow, the empirical evaluation could be expanded. Currently, the benefits of the proposed method are only visible in 5 dimensions ($k=1$ in Fig. 6 - 10). Furthermore, only 1 qualitative example (Fig. 5) regarding the quality of resulting Mapper Graphs does not seem sufficient to establish the performance of the proposed method.

A second weakness that stands out is the non-determinism of Algorithm 7 ("Take a point $p \in S$"). How does this affect the resulting Mapper graph? Are the graphs very different for different random seeds?

---

> ### Author Response · Authors · 2024-12-20
>
> We thank the reviewer for the valuable suggestions, we will incorporate them into the next revision. More specifically:
>
> ## 1. Empirical Evaluation and Benchmarking.
>
> We appreciate the suggestion to include Mapper graph plots for Fashion-MNIST, MNIST, and CIFAR. We will add these plots in the revised version of the paper. Regarding the benchmark plots, we initially planned to evaluate performance for $k$ up to 10 at least. However, these tests were limited to $k \leq 5$ due to memory constraints in our testing environment. Specifically, we encountered "Out of Memory" (OOM) errors when running giotto-tda and kepler-mapper on datasets with higher dimensions. The majority of the tests were feasible up to $k = 7$ only, but not consistently. It is important to highlight that memory usage, in addition to computation time, often poses a significant challenge for classical methods. This limitation will be emphasized in the revised paper. We will report the performances at higher dimensions in the upcoming revision using different settings in order to provide a more comprehensive analysis that would improve the evaluation of scalability.
>
> ## 2. Non-Determinism of Algorithm 7.
>
> Thank you for highlighting the non-determinism in Algorithm 7. We will address this concern in the revision. To mitigate the effects of non-determinism, one possibility is to follow a strategy similar to the case of $\epsilon$-nets: we could select $p \in S$ such that $d(p, Y \setminus S)$ is maximized at each step. However, this would increase the time complexity by introducing a greedy search in every iteration of the main loop. Alternatively, using a fixed random seed would yield reproducible results with no performance drawback.
>
> Regarding the topology of the Mapper graph, the Nerve Theorem states that if an open cover $\mathcal{U}$ of a topological space $X$ forms a \emph{good cover}, i.e. every finite intersection of sets in $\mathcal{U}$ is either empty or contractible, then the nerve $N(\mathcal{U})$ is homotopy equivalent to $X$. This means that in the case of Mapper, that can be seen as a truncation of the Nerve up to dimension 1, the Mapper graph retains topologically relevant information about $X$, which are therefore not dependent on the specific choice of open cover. If this condition is not met, the Mapper graph may no longer reflect the topological features of $X$, and using Mapper in such cases is questionable since the Nerve Theorem does not apply. To conclude, under "good cover" hypethesis, variations in the Mapper graphs are possible due to non-determinism of Algorithm 7, but are constrained by the topology of the underlying dataset and are topologically not relevant. More specifically, in the case of proximity-net, the Mapper graphs are derived from subcovers of the standard cubical cover. Therefore, if the cover for the classical Mapper satisfies the "goodness hypothesis", it must also hold for the subcovers obtained from proximity-net, independently from the choice of the points. In the next revision we will add a discussion about these aspects of Mapper and also evaluate to help the reader with a visual assessment.

---

> ### Comment · Reviewer_qUwr · 2024-12-20
> **Reviewer Response to Authors**
>
> Thank you very much for the detailed answer to my review. The explanations are convincing to me, and if the promised additions, concretely
>   * Mapper graph plots for Fashion-MNIST, MNIST, and CIFAR
>   * performances at higher dimensions and emphasis of limitations
>   * addressing non-determinism and discussing topological (ir)relevance of choice of the points
>
> can be added to the paper, I have no further concerns.
>
> **Update**: The requested additions are in the revision; thank you.

---

### Review · Reviewer_WJji · 2024-12-18

**Summary Of Contributions:**

In this paper, the authors propose a more efficient way to compute an open cover
for the mapper algorithm. Their procedure works by constructing a vp-tree to
allow for efficient neighborhood querries and running their proximiy-net
algorithm afterwards. They afterwards empirically evaluate to which extent this
algorithm is indeed faster then other implementations.

**Audience:**

Yes

**Claims And Evidence:**

Yes

**Requested Changes:**

## Major
See my first critique point
## Minor by descending Priority
- Definition 10: This needs to be rewritten, CP_Y...is a set, so this notation with the iff statement is not useful. I think I understand it, nevertheless I would strongly recommend to use a proper set notation for CP_Y...
- There are two statements about the equivalence of Ball Behavior to the boundless of d(p,q) "hidden" inside the descriptions of figure 4a) and 4b) which are used in Algorithm 4. Even if they may be trivial for the authors, they are used in algorithm 4 and I would explicitly state them before Algorithm 4.
- Algorithm 7: Does it make sense to speak of an "open cover" of Y? For arbitrary proximity functions and arbitrary sets Y, the concept of "openness" is not clear to me. Does maybe "Ensure Cover Y$ may be more accurate here?
- What is dim(Y) in the case of Y not being a subspace of an euclidean space?

**Strengths And Weaknesses:**

## Strengths
+ The paper is easy to follow considering the complexity of the proposed topic. The reader is taken by the hand by the main procedure being split up in small algorithms.
+ The results show that this algorithm is indeed faster then baseline approaches in higher dimensions, making it important for people working in this area
+ Even though I have not checked the proofs of the theorems in detail, the paper seems sound to me. The argumentation and the mathematical constructions and the belonging theorems are looking sound and reasonable.

## Weaknesses
- The paper would benefit from a more depth analysis on how the final mapper graphs look like. Are there outstanding characteristics with respect to the mapper-graph produced with proximity-net compared the standard cover?  So, my request would be to not only have the one visualized example in Figure 5, but a rigorous analysis on how the produced graphs behave.
- I would have been curious to look at the code already to understand how these algorithms are implemented. In 2024, there are various possibilities to share code anonymously....

---

> ### Author Response · Authors · 2024-12-20
>
> We thank the reviewer for their valuable suggestions, which will contribute to improving the quality of the paper.
>
> ## Major Points
>
> 1. We agree with the recommendation to include Mapper graph plots for all the analyzed datasets. These plots will be added in the revised version of the manuscript.
>
> 2. We are exploring ways to share the code anonymously, as it is currently hosted publicly. If the reviewer has suggestions or preferences for a specific platform, we would be happy to accommodate that. Otherwise, we will prepare a zip file containing the entire source code of the Python package, ensuring proper anonymization.
>
> ## Minor Points
>
> 1. We agree with the feedback on notation and will improve it in the next revision.
>
> 2. We will also add references to the figures as comments in the algorithm, as suggested.
>
> 3. We acknowledge the clarification regarding open covers and we will revise the text accordingly. Additionally, throughout the paper, the dimension $dim(Y)$ consistently refers to the doubling dimension, as defined in the preliminary section and applicable to any (pseudo-)metric space.

---

> > ### Comment · Reviewer_WJji · 2025-01-06
> > **Reaction to Authors Updated Version**
> >
> > Dear Authors,
> > thank you for updating the paper and for incorporating my suggestions. The paper has improved. However Major Point 1.) is still not fully satisfied:
> > - All mapper graphs are now PLOTTED, but my request was more about an in-depth analysis of their behavior, what are differing characteristics? Do the densities, node degrees, cluster centrality have specific interesting characteristics, such that they are lower/higher for mapper graphs generated by your algorithm?

---

> > > ### Author Response · Authors · 2025-01-07
> > >
> > > We thank reviewer WJji for pointing out the issue, which was unfortunately overlooked in the previous revision. In the upcoming version, we will include a more thorough analysis of all the datasets considered. By adding these additional metrics, we aim to provide a more quantitative assessment of the quality and characteristics of the produced graphs.

---

> > > > ### Comment · Reviewer_WJji · 2025-01-09
> > > > **Minor Questions Left**
> > > >
> > > > Dear Authors,
> > > > thank you, table 1-3 and the discussion are what I was looking for. I think the result of higher density by lower degrees (which means lower amount of nodes) by your mapper graphs is an interesting finding.
> > > > -The stds are over different nodes, not different runs of your algorithm I guess?
> > > > - Figure 14+16+18: I could not find any mentioning of them in the text, how are they generated? By multiple runs of your algorithm? Could you add a few sentences of elaboration to the paper?

---

> > > > > ### Author Response · Authors · 2025-01-09
> > > > >
> > > > > Thank you for your feedback. The histograms for degree, clustering coefficient, and betweenness centrality are generated by extracting the values associated with each individual node in the plotted Mapper graphs, so each histogram represents a single run. These histograms illustrate the distribution of these metrics across all nodes. The tables provide a summary of these metrics by reporting only the mean and standard deviation (std), calculated across the nodes of the graphs. Additionally, the metrics density and transitivity, being graph-level metrics, are presented without any further breakdown.
> > > > >
> > > > > In the upcoming revision we will clarify the description and add explicit references to the figures.

---

> > > > > > ### Comment · Reviewer_WJji · 2025-01-10
> > > > > > **Reaction to latest Comment**
> > > > > >
> > > > > > Dear Authors,
> > > > > > I am still a little bit confused by the histograms, your y-Axis is labeled "Density". What does that mean? So every bar gives you not an absolute node count but a fraction of nodes? However, then the y-Axis should always be between 0 and 1 (for example not the case for 14c) and all y-values should sum up to 1 (not the case for 14a I would estimate). In the case you have absolult node counts and the y-Label "density" is simply a typo, then 14a) and b) are not useful as the values are <1.
> > > > > > Can you please clarify this?

---

> > > > > > > ### Author Response · Authors · 2025-01-10
> > > > > > > **Clarification about density**
> > > > > > >
> > > > > > > To clarify, by "density" we mean that each histogram has been normalized so that the total area under the histogram equals 1. This is a key distinction between density histograms and probability histograms. In a density histogram, the y-axis values do not represent absolute counts or probabilities directly. Instead, the area of the bars corresponds to relative frequencies, summing to 1 over the entire range. This is why the y-axis can exceed 1, depending on the bin width. For further context, the histograms in question were generated using the following code:
> > > > > > > ```
> > > > > > > sns.histplot(values, kde=True, bins=10, stat='density')
> > > > > > > ```
> > > > > > > This approach ensures that the histograms are density plots, as documented in the Seaborn library ([link to documentation](https://seaborn.pydata.org/generated/seaborn.histplot.html)). If the y-axis were showing probabilities (as in a probability histogram), your observation would be correct: the y-axis values would be between 0 and 1, and the sum of all bar heights would equal 1. We hope this clears up any confusion. Please let us know if further clarification is needed!

---

### Decision · Action_Editor_NFXV · 2025-01-14

**Recommendation:** Accept as is

**Comment:**

All reviewers were generally positive on the submission, and all their concerns have been addressed, with multiple additions that have significantly improved the presentation and the experimental evaluation. All reviewers are unanimous in proposing acceptance for the paper. The only remaining concern is that the implementation has not been checked by any reviewer (as an anonymous link has not been provided). However, this is a minor concern at this point.

**Audience:**

The paper is of interest to a subset of the TMLR audience, in the form of TDA experts and everyone interested in applying TDA analyses in high dimensions.

**Claims And Evidence:**

The paper proposes a more scalable approach for running the Mapper algorithm from the TDA field. The core algorithm is composed of two key improvements in the form of a vp-tree construction procedure, and a proximity-net cover. The mathematical derivation is clear and can be followed easily. The algorithms are described concisely, with a good complexity analysis. Finally, the algorithm is evaluated on several datasets, showcasing good empirical results also through extensive visualizations.